



# Modeling radiative and climatic effects of brown carbon aerosols with the ARPEGE-Climat global climate model

Thomas Drugé[1], Pierre Nabat[1], Marc Mallet[1], Martine Michou[1], Samuel Rémy[2], and Oleg Dubovik[3]

[1]CNRM, Université de Toulouse, Météo-France, CNRS, Toulouse, France
[2]HYGEOS, Lille, France
[3]Université de Lille, CNRS, UMR 8518 – LOA – Laboratoire d'Optique Atmosphérique, Lille, France

**Correspondence:** T. Drugé (thomas.druge@meteo.fr)

**Abstract.** The fraction of organic aerosols, predominantly emitted from biomass burning and biofuel use, that strongly absorbs ultraviolet and short visible light is referred to as brown carbon (BrC). The lifecycle and the optical properties of BrC are still highly uncertain, thus contributing to the uncertainty of the total aerosol radiative effect. This study presents the implementation of BrC aerosols into the atmospheric component of the CNRM climate model and particularly in its aerosol scheme TACTIC,

5   using a BrC parameterization based on the optical properties of Saleh et al. (2014). Several simulations have been carried out with this global climate model, over the period 2000-2014, to analyze the BrC radiative and climatic effects. Model evaluation has been achieved by comparison of single-scattering albedo (SSA), aerosol optical depth (AOD) and absorption aerosol optical depth (AAOD) with AERONET stations at the local scale and with different satellite products at the global scale. This work has mainly shown an improvement, thanks to the BrC implementation and its bleaching parameterization, in total SSA and AAOD,

10   at 440 nm, whether at several AERONET stations or at the regional scale, notably over regions of Africa (AFR) and South America (AME) where large quantities of biomass burning aerosols are emitted. The annual global BrC effective radiative forcing (all-sky conditions) has been calculated in terms of aerosol-radiation interactions (ERFari, $0.029 \pm 0.006$ W m$^{-2}$) and in terms of aerosol–cloud interactions (ERFaci, - $0.024 \pm 0.066$ W m$^{-2}$). Over the AFR and AME regions, the study shows respectively a positive ERFari of $0.292 \pm 0.034$ and of $0.085 \pm 0.032$ W m$^{-2}$ on annual average, close to the BrC radiative

15   effect calculated in other studies. This work also shows that the inclusion of BrC causes a statistically significant low-level cloud fraction increase over the South-East Atlantic Ocean during the burning season caused in part by a vertical velocity decrease at 700 hPa (semi-direct aerosol effect). Lastly, this study also highlights that the low-level cloud fraction changes, associated with more absorbing biomass burning aerosols when BrC aerosols are included, contribute to an increase, at 700 hPa, of both solar heating rate and air temperature over this region.



# 1 Introduction

The representation of aerosols is still a fairly large source of uncertainty for climate models (Myhre et al., 2013; Szopa et al., 2021). In addition to affecting cloud properties and precipitation patterns (first and second indirect effects), some aerosols such as black carbon (BC) particles warm the atmosphere by directly absorbing solar radiation while others, such as sea-salt or

nitrate particles, tend to scatter it, that leads to cooling of the atmosphere (Haywood and Boucher, 2000; Bond et al., 2013; Zhang et al., 2020). In modeling studies, organic aerosols (OA, also referred as organic matter, OM), which add primary organic aerosols (POA) and secondary organic aerosols (SOA), are usually considered to be strongly scattering (Myhre et al., 2013) but recent works have shown that a part of the OA, known as brown carbon (BrC), can absorb ultraviolet (UV) and short visible light, predominately at near-UV wavelengths (Kirchstetter et al., 2004; Yang et al., 2009; Hecobian et al., 2010; Arola et al.,

2011; Kirchstetter and Thatcher, 2012).

Primary BrC has primarily been associated with biomass burning (BB) and biofuel (BF) combustion (Andreae and Gelencsér, 2006; Desyaterik et al., 2013; Feng et al., 2013; Washenfelder et al., 2015). This type of combustion also emits BC as well as non-absorbent organic carbon (OC), which makes the BrC optical properties determination particularly difficult (Wang et al., 2018). Secondary BrC can be produced from the photo-oxidation of volatile organic compounds (Jacobson, 1999; Nakayama

et al., 2012; Sareen et al., 2013; Laskin et al., 2014) and also from aqueous-phase reactions in droplets (Updyke et al., 2012; Nguyen et al., 2012). Other secondary BrC sources exist, such as homogeneous and heterogeneous reactions of catechol or phenolic compounds (Pillar et al., 2014; Smith et al., 2016; Yu et al., 2016; Lavi et al., 2017; Pillar and Guzman, 2017). However, BrC from BF and evenmore BrC from BB contribute more to solar radiation absorption than that from the other sources (Chakrabarty et al., 2010; Kirchstetter and Thatcher, 2012; Saleh et al., 2014). The BrC absorption is also affected by

the combustion efficiency (Chen and Bond, 2010; Saleh et al., 2014; Pokhrel et al., 2016). It is therefore a function of the BC-to-OA ratio in the emissions, that is dependent on the emission source burning conditions. Akagi et al. (2011) have shown that a high BC-to-OA ratio, corresponding to a fast and hot fire such as a savannah fire, is correlated with a strong BrC absorption. Conversely, a low BrC absorption will be due to a small BC-to-OA ratio, corresponding to a slower and smoldering fire. Once BrC particles are emitted into the atmosphere, their chemical composition changes with aging through different processes (Lee

et al., 2014; Zhong and Jang, 2014; Forrister et al., 2015; Zhao et al., 2015). BrC can be photolyzed and degraded to be less absorbing when directly exposed to solar radiation (i.e., bleaching). This phenomenon is source dependent, the higher molar weight and less-volatile BrC being more resistant to bleaching (Wong et al., 2017). This BrC bleaching, which can reduce its radiative effect up to 50% (lower absorption), takes place between a few minutes to one day after its emission (Zhong and Jang, 2011; Lee et al., 2014; Forrister et al., 2015; Zhao et al., 2015; Wang et al., 2016; Brown et al., 2018). In addition, recent

laboratory studies have revealed the formation of secondary BrC by certain chromophores through photochemical reactions in the aqueous phase, thereby photo-enhancing the particle brownness (Lambe et al., 2013). Further studies are still needed to better understand the BrC photo-enhancement and bleaching, in order to accurately quantify the timescale, species-dependency, and impacts of these two compensating processes.



Several modeling studies have attempted to simulate BrC in global models and to estimate its radiative forcing. The effective radiative forcing (ERF) is a useful measure of defining the impact on the Earth's energy imbalance of a radiative anthropogenic or natural perturbation (Myhre et al., 2013; Forster et al., 2016; Smith et al., 2020). This concept and its calculation, that will be used in this study, are described in details in section 3.3. Several alternative indicators of radiative forcing appear in the litera-

ture. The difference between ERF and radiative forcing (RF) is that ERF includes all rapid adjustments (including tropospheric and land surface ones), whereas RF only includes adjustments due to stratospheric temperature changes (Sherwood et al., 2015; Myhre et al., 2013; Smith et al., 2020). Unlike the ERF and RF, the instantaneous radiative forcing (IRF), also called radiative effect (RE), corresponds to the initial perturbation to the Earth radiation budget and does not include adjustments (Smith et al., 2020). When the direct radiative effect of an aerosol is calculated between two different climate states, it is usually called direct

radiative forcing (DRF) of this aerosol. A few studies, based partly on global chemical transport models (CTMs) combined with radiative transfer models, have simulated BrC IRFs (Park et al., 2010; Wang et al., 2014; Saleh et al., 2015; Jo et al., 2016; Brown et al., 2018; Wang et al., 2018; Tuccella et al., 2020; Liu et al., 2020) or BrC DRFs (Feng et al., 2013; Lin et al., 2014; Wang et al., 2014) ranging from +0.04 to +0.57 W m$^{-2}$ at the Top of the Atmosphere (TOA). This wide range is due to the lack of knowledge about BrC sources, ageing processes and optical properties but also to the diversity of BrC implementations

in atmospheric models.

This diversity can be grouped into two main categories. The first approach consists in assuming that BrC corresponds to a fraction of OA and in attributing to BrC specific optical properties (Feng et al., 2013; Lin et al., 2014; Wang et al., 2014; Tuccella et al., 2020). This method is relatively simple to implement, however, the assumed BrC optical properties based on laboratory measurements are not well constrained (Wang et al., 2014). In order to overcome this limitation, some studies used

both the lower and the higher bounds from laboratory studies (Feng et al., 2013; Lin et al., 2014). Other uncertainties and limitations include the fraction of OA that can be considered as BrC and the variation of this fraction according to the source. In their study, Feng et al. (2013) assumed that BrC corresponds to 66% of POA from BF and BB emissions. Lin et al. (2014) made the assumption that all BF/BB POA and all biogenic/anthropogenic SOA are BrC. In Wang et al. (2014) and Tuccella et al. (2020) studies, BrC corresponds, respectively, to 50% and 25% of the BF and BB POA, and to aromatic SOA. In the

case of strong (moderate) BrC absorption assumptions, Feng et al. (2013) showed a BrC DRF of +0.11 W m$^{-2}$ (+0.04 W m$^{-2}$) at the top of the atmosphere with the IMPACT (Integrated Massively Parallel Atmospheric Chemical Transport) CTM and attributed 19% of the anthropogenic aerosol absorption to BrC. Also with the IMPACT model, Lin et al. (2014) estimated in their study a BrC IRF between +0.22 and +0.57 W m$^{-2}$, which corresponds to between 27 and 70% of the BC absorption. Finally, with the GEOS-Chem global CTM, Wang et al. (2014) and Tuccella et al. (2020) estimated a BrC IRF of +0.11 and of

0.27 W m$^{-2}$, respectively.

The second approach in implementing BrC in climate models is to parameterize the imaginary refractive index of BrC according to an independent variable such as the modified combustion efficiency (MCE), that is a function of the CO-to-CO$_2$ ratio in the emissions (Jo et al., 2016), or such as the BC-to-OA ratio in the emissions (Park et al., 2010; Saleh et al., 2015; Brown et al., 2018; Wang et al., 2018). This allows BrC properties to be tied to burning conditions and to better represent

the spatial and temporal variabilities of the BrC absorption. In their GEOS-Chem global CTM Saleh et al. (2015) considered





that BrC corresponds to 100% of the BB and BF emissions and modified the imaginary part of its refractive index based on the BC-to-OA ratio (Saleh et al., 2014). They estimated a global mean effect of OA absorption of +0.12 W m$^{-2}$ when BrC is internally mixed and of +0.22 W m$^{-2}$ with an external mixture. The Brown et al. (2018) study carried out with the CAM5 (Community Atmosphere Model version 5) model, which includes the Saleh et al. (2014) parameterization in addition to a

BrC bleaching parameterization that ages BrC to 25% of its original absorption over about 1 day, showed a global ERF due to aerosol-radiation interactions (ERFari) of +0.13 W m$^{-2}$ without BrC bleaching effects and of +0.06 W m$^{-2}$ with the BrC bleaching parameterization. With the same parameterization but considering all OA as BrC and using a constant BC-to-OA ratio for each source (0.05 for BB and 0.12 for BF), Wang et al. (2018) estimated, with the GEOS-Chem global CTM, a global IRF of +0.048 W m$^{-2}$ with BrC bleaching effects. Finally, the BrC effects on clouds and atmospheric dynamic have only

rarely been addressed in past studies. However, Brown et al. (2018) showed a global annual effective radiative forcing due to aerosol-cloud interaction (ERFaci) of 0.01 W m$^{-2}$.

In the present study, we implemented BrC as a new prognostic aerosol, in addition to OA and BC, into the aerosol scheme TACTIC of the CNRM (National Centre for Meteorological Research) global climate atmospheric model called ARPEGE-Climat (Roehrig et al., 2020) and studied its radiative (ERFari and ERFaci) and climatic effects over the period 2000-2014.

To compute BrC optical properties, we used the Saleh et al. (2014) imaginary refractive index with a constant BC-to-OA ratio as well as a bleaching parameterization. The climate model and its aerosol scheme description, the BrC implementation as well as the experimental setup are described in Sect. 2 and 3. Sect. 4 presents the model results of this study with, in a first step, the evaluation of the new aerosol scheme at the local scale with Aerosol Robotic Network (AERONET) data and at the global scale with original satellites products, and, in a second step, the BrC radiative and climatic effects. Lastly, conclusions

are summarized in Sect. 5.

## 2 Model description

### 2.1 The ARPEGE-Climat global climate model

The ARPEGE-Climat global spectral model, used in this study, is the atmospheric component of the Centre National de Recherches Météorologiques (CNRM) climate models. It is used here in a similar version as the one described in detail in

Roehrig et al. (2020). The ARPEGE-Climat model consists, as well as other atmospheric models, of a dry dynamical core and a suite of physical parameterizations, that represent diabatic processes. The atmospheric physics and dynamics are computed using a spectral transform on the sphere operating at a T127 triangular grid truncation that is equivalent to a spatial resolution of about 150 km in both longitude and latitude, as illustrated in Figure 1. ARPEGE-Climat is a "high-top" model, with 91 vertical levels from the surface to 0.01 hPa in the mesosphere. Concerning the radiation scheme, ARPEGE-Climat uses a long-wave

(LW) radiation scheme based on the rapid radiation transfer model (RRTM, Mlawer et al. 1997) and the FMR short-wave (SW) radiation scheme with six spectral bands (Fouquart et al., 1980; Morcrette et al., 2008).

The ARPEGE-Climat global climate model includes the SURFace EXternalisée (SURFEX) modeling platform in its version 8 to simulate surface state variables and fluxes at the Earth's surface (Decharme et al., 2019). Over the land surface, the





Interaction Soil-Biosphere-Atmosphere (ISBA, Noilhan and Mahfouf 1996) land surface model coupled to the Total Runoff Integrating Pathways (CTRIP, Decharme et al. 2019; Voldoire et al. 2019) river model are used to represent physical processes. Lastly, the ARPEGE-Climat model includes an interactive aerosol scheme described thereafter.

## 2.2 The TACTIC aerosol scheme

TACTIC (Tropospheric Aerosols for ClimaTe In CNRM) is the bulk-bin aerosol scheme used in the climate models of CNRM (Michou et al., 2015; Nabat et al., 2015a), originally derived from the ECMWF IFS aerosol module (Morcrette et al., 2009; Rémy et al., 2019), and representing the main tropospheric aerosol types and their interactions with climate. The version used in the present study is based on the one used in the CNRM-ESM2-1 simulations (Séférian et al., 2019) carried out for the sixth phase of the Coupled Model Intercomparison Project (CMIP6), and described in detail in Michou et al. (2020). Compared to

the latter, TACTIC here includes the representation of nitrate and ammonium particles as described by Drugé et al. (2019), modifications on sea-salt emissions described in Nabat et al. (2020), as well as further developments described thereafter concerning the formation of sulfate particles, the aerosol wet deposition and the aerosol-radiation coupling.

In summary, the TACTIC aerosol scheme simulates the physical evolution of seven aerosol types that are supposed externally mixed: desert dust, sea salt, black carbon, organic matter, sulfate and recently added nitrate and ammonium particles. Terrestrial

biogenic SOA are not formed explicitly but are taken into account through the climatology of Dentener et al. (2006) while oceanic biogenic SOA and aromatic SOA are not yet considered. To represent the particle size spectrum, the TACTIC aerosol scheme includes 16 prognostic variables or aerosol bins: three size bins are used for desert dust (DD, respective limit diameters of the three bins are 0.01 to 1.0, 1.0 to 2.5 and 2.5 to 20 $\mu$m) and sea salt (SS, respective limit diameters of the three bins are 0.01 to 1.0, 1.0 to 10.0 and 10.0 to 100.0 $\mu$m), two bins separating hydrophilic and hydrophobic particles for organic matter

(OA) and for black carbon (BC), one size bin for sulfate ($SO_4$) particles and for sulfate precursors, notably sulphur dioxide ($SO_2$). Finally, nitrate particles ($NO_3$) are divided into two bins (for gas-to-particle reactions and for heterogeneous chemistry) and the last two tracers are used for ammonium ($NH_4$) and ammonia ($NH_3$). Aerosols can be interactively emitted from the surface (DD or SS) as a function of surface wind and soil characteristics or the scheme can consider external emission datasets, including those for anthropogenic and/or biomass burning particles (BC, OA, $SO_2$, and $NH_3$). As described in Michou et al.

(2015), a multiplier coefficient of 1.5, based on analysis of fresh urban emissions (Turpin and Lim, 2001), is applied to organic carbon emissions in order to take into account the conversion of organic carbon into organic matter.

In TACTIC, the formation of sulfate was originally based on the conversion of sulfate precursors (summarized as $SO_2$) into sulfate assuming an exponential decay with a time constant depending on the latitude (Rémy et al., 2019). In the present version, the sulfate formation now deals explicitly with the chemical oxidation of sulfate precursors into sulfate. Three oxidants

are taken into account: OH in the gas phase, and $H_2O_2$ and $O_3$ in the aqueous phase. The chemical mechanism is derived from that of Berglen et al. (2004), with updated reaction rates (Seinfeld and Pandis, 2006; Burkholder et al., 2015), and updated Henry's law solubility coefficients (Sander, 2015). The concentrations of the oxidants consist in monthly climatologies built from diagnostics of the CAMSiRA global reanalysis of atmospheric composition (Flemming et al., 2017).





The modifications in the aerosol wet deposition scheme consist in considering together the sum of large-scale and convective precipitation to scavenge aerosols with this resulting precipitation flux, and in refining the representation of the in-cloud scavenging according to the type of cloud. Indeed, the in-cloud scavenging, which represents most of the total scavenging, is not the same for liquid, mixed-phase and solid clouds. TACTIC now uses specific coefficients for these three types of clouds,

based on those proposed by Bourgeois and Bey (2011). The resuspension of aerosols when precipitation evaporates has also been improved, using a correction factor described in de Bruine et al. (2018). Finally a mass fixer is now applied to ensure conservative tracer transport (Bermejo and Conde, 2002).

The atmospheric model represents the interactions between particles and radiation (aerosol direct effect) and between particles and cloud albedo (first aerosol indirect effect, see Michou et al., 2020 for details). On the other hand, the second indirect

aerosol effect, that corresponds to interactions between aerosols and cloud precipitation, is not included for the time being. With regards to aerosol-radiation interactions, TACTIC is able to produce different aerosol optical properties (extinction, SSA and asymmetry parameter, see Table A1) for different wavelengths, based on look-up tables pre-calculated using a Mie code and the aerosol sphericity hypothesis (Ackerman and Toon, 1981), depending on the relative humidity, except for DD and hydrophobic BC and OA. In this version of ARPEGE-Climat, the interaction between aerosols and radiation has been improved,

the radiative code is provided with all aerosol optical properties for each aerosol bin and each wavelength both in the shortwave and longwave spectra.

Finally, it is worth mentioning that the TACTIC aerosol scheme keeps a reasonable computation cost, notably thanks to several simplifications like not calculating on-line the aerosol optical properties or not interacting with gaseous tropospheric chemistry.

## 20  2.3  Brown carbon implementation

A new prognostic aerosol species has been added into the TACTIC aerosol scheme: the BrC. The first step of this implementation, that allows a good representation of the spatial and temporal variability of the BrC absorption, consisted in separating BrC from OA according to their sources. To do this, OA emissions were separated into three sources: biomass burning (BB), fossil fuel (FF) and biofuel (BF) as presented in Figure 2. In the model BB emissions, provided over the period 2000-2014, are

those described in Van Marle et al. (2017) with the GFED4s (Global Fire Emissions Database) as an anchor dataset. Then, BF emissions mainly coming from the residential, industry and energy sectors, and FF emissions, are those of the CEDS (Community Emissions Data System) inventory released for CMIP6 (Hoesly et al., 2018). In this parameterization, BrC corresponds to what is emitted by BB and BF while OA corresponds to what is only emitted by FF.

We thus consider OA as a non absorbing aerosol, as shown by most observations (Laskin et al., 2015), while BrC is consid-

ered as an absorbing aerosol. The second step of this implementation work was therefore to calculate the BrC optical properties at different wavelengths and different relative humidity using a Mie code (Toon and Ackerman, 1981). For this purpose, we used the BrC refractive index ($RI_{BrC}$) shown in eq. 1: the real part (1.53) is the one commonly used in previous studies (Chen and Bond, 2010; Arola et al., 2011; Lin et al., 2014; Tuccella et al., 2020) and remains close to that used in the model for OA





(1.45) while the imaginary part comes from experimental results (Saleh et al., 2014):

$$RI_{BrC} = 1.53 + 0.016.log_{10}(BC - to - OA) + 0.04.(550/\lambda)^{\omega}i \tag{1}$$

$$\omega = \frac{0.21}{\text{BC-to-OA+0.07}} \tag{2}$$

The imaginary part depends on the BC-to-OA ratio from BB and BF emissions. Its wavelength ($\lambda$) dependence is further detailed in Eq. (2). As in other climate models, we chose to use a constant global BC-to-OA emission ratio. It is therefore important to note that this assumption does not reflect all burning conditions and specific fires. Wang et al. (2018) fixed a

global average BC-to-OA emission ratio for each source (0.12 for BF and 0.05 for BB) and indicate that the variability of the BC-to-OA ratio (0-0.23 for BF and 0.03-0.06 for BB in the GFED4s emission inventory) is underestimated because all burning conditions are not represented. This is reinforced by Ramo et al. (2021) who show that, since the 1990s on sub-Saharan Africa, burned area and fire carbon emissions are underestimated because they are strongly impacted by small fires, which are undetected by coarse resolution satellite data. Brown et al. (2018) ran a sensitivity experiment with a BC-to-OA ratio set to

0.08, and based on their results we adopted this ratio. The BrC geometric median diameter is assumed to be of 0.1 μm with a standard deviation of 1.6. The density of BrC, usually ranging from 1 to 1.5 (Feng et al., 2013; Lin et al., 2014; Shamjad et al., 2016; Wang et al., 2018; Tuccella et al., 2020), is assumed to be of 1 g cm$^{-3}$ as in the study of Brown et al. (2018).

We implemented a BrC bleaching parameterization represented by the passage from an hydrophobic bin (fresh bin) to an hydrophilic bin (aged bin), each having specific optical properties, the aged bin absorption being lower than that of the fresh

bin. The passage is done with a characteristic time of 1 day. This characteristic time ranges from a few minutes to several days in the literature (Zhong and Jang, 2011; Lee et al., 2014; Forrister et al., 2015; Zhao et al., 2015; Wang et al., 2016; Vakkari et al., 2018; Zhang et al., 2020), but 1 day is commonly used (Wang et al., 2016; Brown et al., 2018; Wang et al., 2018). Then, knowing that the level of BrC absorption decrease over time reaches up to 75% of the initial absorption in certain studies (Wang et al., 2018; Zhang et al., 2020), we tested several absorption decreases (25, 50 and 75%). The most satisfactory results

were obtained with the averaged value (50%), therefore, and for clarity reasons, we decided to show here only results with the 50% value. The aerosol scheme can also be ran without BrC aging. In this case, BrC is only represented by one hydrophilic variable. The different BrC optical properties are summarized, at 350 and 550 nm, in Table A1 for a relative humidity of 0 and 80%. A BrC dry deposition velocity of 0.1 cm s$^{-1}$, close to the first SS and DD bin deposition velocity (Michou et al., 2015), is set over all surfaces (ocean, sea ice, land and land-ice). Lastly, the efficiency with which BrC particles are washed out is

0.001 for rain and 0.01 for snow (below-cloud scavenging), while the fraction of BrC included in a cloud droplet is 0.25 for liquid clouds, and 0.06 for mixed-phase and ice clouds (in-cloud scavenging). For reasons of consistency and awaiting further studies, these values are the same as those used for OA aerosols.





One limitation of this study is to neglect absorption by biogenic (Lin et al., 2014; Saleh et al., 2015) and aromatic SOA (Wang et al., 2014; Jo et al., 2016; Wang et al., 2018). However, some studies have shown that the absorption of the primary BrC from BB and BF emissions usually dominates that of the absorbing SOA (Martinsson et al., 2015; Laskin et al., 2015).

## 3 Methodology

### 3.1 Reference datasets

In this study, different datasets have been used to evaluate the ability of the ARPEGE-Climat model to reproduce the total absorption aerosol optical depth (AAOD), single-scattering albedo (SSA) and aerosol optical depth (AOD). First, given their spatial and temporal scales, four satellite products have been used to provide estimates of the total AAOD and SSA (at 440 nm) and of the total AOD (at 550 nm):

- PARASOL-GRASP (2006-2012, 1°resolution, Chen et al. 2020) for the AAOD and SSA at 440 nm and the AOD at 550 nm. This satellite product, initially developed by the science team at LOA (Laboratoire d'Optique Atmosphérique, Lille, France), is obtained by the Generalized Retrieval of Atmosphere and Surface Properties (GRASP) algorithm from POLDER-PARASOL observations. The GRASP algorithm, described in Dubovik and King (2000); Dubovik et al. (2011, 2014), was developed for deriving extensive aerosol properties from a variety of remote sensing instruments. In this study, version 2.1 L3 of the PARASOL-GRASP «models» configuration is used. The AAOD or AOD uncertainty does not exceed 0.01 (0.02 for SSA and AOD over ocean) at all wavelengths (Chen et al., 2020). SSA values are aggregated only when AOD (443 nm) is greater or equal to 0.3 over land and 0.02 over ocean. This very low threshold (0.02) for filtering SSA over ocean was chosen in order to retain a sufficient number of SSA and AOD retrievals (Chen et al., 2020).

- OMI (2005-2019, 1°resolution, Levelt et al. 2006; Torres et al. 2007) for the AAOD and SSA (440 nm) and the AOD (550 nm). The OMI (Ozone Monitoring Instrument) dataset comes from a spectrometer aboard NASA's Earth Observing System's Aura satellite and is archived at the NASA Goddard Earth Sciences Data and Information Services Center (Ahmad et al., 2003; Jethva et al., 2014). In this study, the OMI data Level 3 version 1.7.2 is used. AAOD and SSA uncertainties are respectively estimated at about 0.01 and 0.03.

- MACv2 (2001-2016 with the reference year 2005, directory spectral/ssp_31bands, 1°resolution, Kinne 2019) for the AAOD and SSA (440 nm) and the AOD (550 nm). The Max Planck Institute Aerosol Climatology (MACv2) is an update of the MAC-v1 climatology described in Kinne et al. (2013). This data set provides monthly aerosol optical properties derived from a combination of observations, like those from the AERONET and MAN (Marine Aerosol Network, Smirnov et al. 2009) ground networks, and model outputs derived from the AeroCom global modeling initiative (Kinne et al., 2006; Koffi et al., 2016). The inter-annual variability is taken into account through an AOD change for the anthropogenic aerosols but only monthly variations are used for natural aerosols. The AAOD uncertainty is estimated at about 0.003.

- FMI_SAT (1995-2017, 1°resolution, Sogacheva et al. 2020) for the AOD at 550 nm. FMI_SAT is the name given to the merged AOD product presented in Sogacheva et al. (2020). This product, which provides AOD monthly data, was built from





12 individual satellite products and evaluated with the AERONET ground network. It provides AOD data with an uncertainty reaching 0.006 on average and up to 0.05 in regions with high AOD (Sogacheva et al., 2020).

Second, we considered Aerosol Robotic Network (AERONET, from 2000 to 2020 depending on the station, Holben et al. 1998) data. This network consists in globally distributed ground-based sun photometers which provide local column-integrated aerosol properties at different solar wavelengths, including 440 nm. The column extinction Ångström exponent can be directly calculated from the wavelength-dependent AOD measurements (Eck et al., 1999). AAOD and SSA, which are provided only when AOD $\geq$ 0.4, are then calculated with algorithms utilizing both the spectral AOD and the spectral angular distribution of the sky radiances (Dubovik et al., 2000; O'neill et al., 2003). In this study, monthly average quality-assured data (level 2.0, version 3 described in Sinyuk et al. 2020) have been used. For comparison to our model results, AAOD and SSA data at 440 nm were directly used. AOD data have been calculated at 550 nm using Ångström coefficients between the closest available upper and lower wavelengths. AERONET uncertainties have been reported in several papers. Eck et al. (1999) and Kinne et al. (2013) indicate that the AOD and AAOD uncertainties are approximately 0.01. Concerning the SSA, Dubovik et al. (2000) report an uncertainty of 0.03. Eighteen stations have been selected because they reported at least eight daily data to derive monthly means and we could compute at least 3 monthly values over 2000-2020 for a given month, and also because of their location. We retained 6 sites in Africa, 3 in South America and 9 in Europe and Asia as shown in Figure 1 (see also Table 1).

It is important to mention that the satellite datasets, as well as the AERONET data, were obtained during day time only, contrary to the model data, which were obtained over the whole day (night plus day). The main characteristics of the reference datasets described here are summarized in Table 2.

## 3.2  Simulations

Two main configurations of the aerosol scheme have been used in this study, including or not BrC aerosols. All simulations consist in AMIP-type simulations with prescribed monthly sea surface temperature- (SST) and sea ice fraction, cover 15 years from 2000 to 2014 and include two members, so the total number of simulated years is of 30. The simulation defined as the baseline for this work, using ARPEGE-Climat without the BrC parameterization, is called NOBRC. The second simulation, called BRC, differs from NOBRC as it considers a new BrC tracer as well as a bleaching parameterization. For a sensitivity study an additional simulation, named BRC_NOBL, has been performed. This third simulation considers the new BrC tracer but does not take into account the bleaching parameterization. The main characteristics of these three simulations are summarized in Table 3.

For information multiplier coefficients of 1.7 (simulations with BrC) or of 1.8 (simulations without BrC), in addition to the first multiplier coefficient of 1.5 presented in section 2.2, have been applied to particulate aerosol (OA and BC) biomass burning emissions, as done in other studies (e.g., Kaiser et al. (2012)). These coefficients (1.7 and 1.8) are based on an AOD (550 nm) comparison between that simulated by the model (2000-2014) and that provided by the merged AOD product FMI_SAT (1995-2017, described below) over regions influenced by large biomass burning emissions (10 - 40°E / 0 - 15°S over Africa and 40 - 70°W / 0 - 20°S over South America). This comparison was performed over the months of July, August and September (JAS),



which is the period with the most intense biomass burning activity over the tropics. The objective of these coefficients was to ensure similar regional JAS total AOD between simulations and FMI_SAT.

## 3.3 Effective radiative forcing calculation

A forcing concept, that allows all physical variables to respond to perturbations except those about the ocean and sea ice, has
been introduced by Myhre et al. (2013) and Shindell et al. (2013): the effective radiative forcing (ERF). In order to estimate the BrC ERF, we used the method recommended in Ghan (2013). In this method, the total ERF can be differentiated between aerosol-radiation interactions (ERFari), aerosol-cloud interactions (ERFaci) and a residual term representing mainly surface-albedo changes (ERFres):

$$ERF = \Delta(F) = ERFari + ERFaci + ERFres \qquad (3)$$

$$ERFari = \Delta(F - F_{clean}) \qquad (4)$$

$$ERFaci = \Delta(F_{clean} - F_{clear,clean}) \qquad (5)$$

In Eq. (4), $\Delta$ refers to the difference between the simulation with (BRC) and the simulation without (NOBRC) BrC aerosols. The F variable represents the TOA net radiation flux (SW + LW), $F_{clean}$ refers to the TOA net radiation flux (SW + LW) neglecting both aerosol scattering and absorption. Then, in Eq. (5), $F_{clear,clean}$ is the clear-sky fluxes when neglecting both aerosol scattering and absorption. Other methods of calculating ERFari and ERFaci exist but the Ghan (2013) technique seems to be particularly accurate (Zelinka et al., 2014). It is important to note here that the BrC indirect effect is taken into account in
the same way as the OA aerosol indirect effect.

## 4  Model results

### 4.1  Evaluation of the aerosol scheme

#### 4.1.1  Local scale

Firstly, ground-based AERONET observations are used to locally evaluate the ARPEGE-Climat simulations. Figure 3, Figure 4
and Figure 5 present, respectively, the SSA (440 nm, 400 nm for MACv2), the AOD (550 nm, 565 nm for PARASOL-GRASP)





and the AAOD (440 nm, 400 nm for MACv2) annual cycles at the selected AERONET sites (linear interpolation at the station point), simulated by the ARPEGE-Climat model (NOBRC, BRC_NOBL and BRC simulations), and retrieved by AERONET as well as by our other reference products (PARASOL-GRASP, OMI, MACv2 and FMI_SAT). The 550 nm wavelength was chosen for the AOD evaluation in order to have in hand better reference data and in particular the FMI_SAT product. It is

interesting to note here that the different satellite products (see Table 2 for reference datasets details) are sometimes not very consistent with the AERONET data, both in annual cycle and in annual average. Over regions with high biomass burning activity such as Southern Africa and Southern America, Figures 3 and 5 indicate a decrease (increase) in SSA (AAOD) for simulations including BrC at all AERONET stations, particularly during the JAS period. These Figures also show that these changes are even more pronounced with BRC_NOBL. Conversely, Figure 4 shows a very little BrC impact on total aerosol

AOD; indeed, the three simulations show similar values at all AERONET stations. In more detail, Figure 3 shows SSA BRC and BRC_NOBL SSA in better agreement with the different observation datasets at all stations over Africa and South America than NOBRC SSA. The bleaching parameterization allows, at some AERONET stations, to better represent the observations (e.g. HESS, Windpoort or Skukusa). However, at other AERONET stations (e.g., Lubango, Ascension_Island and Mongu Inn), the BRC_NOBL simulation seems to show results in better agreement with the observations. At Windpoort or Skukusa, Figure 3

shows a SSA decrease of about 0.05 with the BRC simulation during the summer period which is consistent with the SSA decreases shown by all observation datasets. At Mongu Inn, there is a stronger SSA decrease of 0.08, associated to an AAOD increase of 0.06 over the JAS period between the NOBRC and BRC_NOBL simulations. BRC_NOBL is therefore the closest to the AERONET observations. As for the SSA, the BrC implementation allows to simulate AAOD close to all observations. Indeed, Figure 5 shows that AAOD is systematically underestimated with the NOBRC simulation, both over Africa and South

America stations. Nevertheless, when the bleaching parameterization is not taken into account (BRC_NOBL), AAOD are often slightly overestimated (up to 0.03 during JAS) compared to all observations. On the other hand, the BRC simulation shows lower AAOD values than the BRC_NOBL simulation, and this sometimes results in underestimated AAODs at some sites (e.g., Lubango or Mongu Inn). Over Europe and Asia (AERONET stations grouped under "Other"), Figures 3 and 5 show that the BrC implementation has a very little impact on the total SSA and AAOD as in these regions the BrC AOD represents less

than 7% of the total aerosol AOD. However, Figures 3 and 5 show with the BrC implementation a SSA decrease and an AAOD increase at some Asian stations such as Jabiru (northern Australia) (-0.05 for SSA and +0.01 for AAOD during summer and fall), or Beijing and XiangHe (China), with an AAOD increase (+0.01) during spring.

   For the sake of clarity, SSA, AOD and AAOD averages over the JAS period are summarized in Table 4. On average over all African AERONET stations, the BRC simulation presents a mean SSA equal to $0.890 \pm 0.002$ which is in better agreement

with the range of the observations ($0.876 \pm 0.007$ - $0.904 \pm 0.003$) than that of the NOBRC simulation ($0.918 \pm 0.001$) and that of the BRC_NOBL simulation ($0.865 \pm 0.002$). Over these stations, the total aerosol AAOD simulated by NOBRC is equal to $0.028 \pm 0.001$ against $0.040 \pm 0.002$ for the BRC and $0.050 \pm 0.002$ for the BRC_NOBL simulations. Compared to the different reference datasets showing an AAOD between $0.039 \pm 0.002$ and $0.059 \pm 0.003$, the NOBRC simulation therefore underestimates observations in contrast to the BRC and BRC_NOBL simulations. Table 4 shows similar results for

the South America AERONET stations with a simulated SSA ($0.917 \pm 0.008$ and $0.900 \pm 0.008$) and AAOD ($0.045 \pm 0.012$





and 0.055 ± 0.012) with the BRC and BRC_NOBL simulations respectively that fall within the range of the different reference datasets (0.873 ± 0.001 - 0.932 ± 0.004 for SSA and 0.033 ± 0.008 - 0.068 ± 0.010 for AAOD) contrary to the NOBRC simulation (0.948 ± 0.004 for SSA and 0.027 ± 0.006 for AAOD). Over the Africa AERONET stations, Table 4 indicates that AOD from BRC and BRC_NOBL are slightly higher (0.300 ± 0.014 and 0.291 ± 0.013 respectively) than our reference

range (0.205 ± 0.008 - 0.279 ± 0.015). However, over South America AERONET stations, BRC and BRC_NOBL AOD are within this range. Finally, as previously discussed, Table 4 shows only little differences between the three simulations at the European/Asian AERONET stations except for slightly lower SSA and slightly higher AAOD for the BRC and BRC_NOBL simulations, closer then to the reference datasets.

Biases and normalized root mean square error (NRMSE, defined as the ratio between the root mean square error (RMSE) and

the average of the AERONET data) boxplots between observed (average of the Africa and South America AERONET stations) and predicted SSA, AOD and AAOD are presented in Figures 6 and 7. These Figures clearly show a decrease in bias and NRMSE of SSA and AAOD with the BrC implementation, both in annual and JAS statistics. The most marked improvement occurs when the bleaching parameterization is taken into account (BRC simulation): the median SSA bias is reduced from 0.025 (annual) and 0.035 (JAS) in the NOBRC simulation to almost zero in the BrC simulation. If the bleaching is not taken

into account (BRC_NOBL simulation), the median SSA bias is of - 0.015 in annual and of - 0.020 over JAS. Figure 6 also shows similar results for the AAOD with a strong reduction of the bias with the BRC simulation to reach very low values both in annual and JAS statistics. However, Figures 6 and Figure 7 show a slight increase of the median AOD bias (+ 0.010 in annual and + 0.020 over JAS) and NRMSE (+ 0.100 in annual and + 0.050 over JAS) with the BrC implementation. The different diagnostics presented in this section therefore show a significant SSA and AAOD improvement thanks to the BrC

parameterization and this betterment is even clearer when the BrC bleaching parameterization is also taken into account. For clarity reasons, only the NOBRC and BRC simulations will therefore be presented in the remainder of this study.

### 4.1.2 Global scale

Simulated SSA, AOD or AAOD from the two model simulations, without BrC and with BrC and its bleaching parameterization are now compared in a climatologicaly perspective to several monthly satellite products. We performed these comparisons

globally and over regions influenced by large biomass burning emissions, namely the AFR region (AFRica, 15°W - 40°E / 0 - 25°S) and the AME region (South AMErica, 70°- 20°W / 0 - 25°S) (see Figure 1). Figure 8 presents the comparison of the SSA at 440 nm simulated by the model (NOBRC and BRC) with that retrieved from the MACv2, PARASOL-GRASP and OMI datasets over the year and over JAS. All temporal and spatial averages are summarized in Table 5. The SSA clearly decreases when considering the BrC aerosol, especially during the JAS season over AFR (-0.024) and AME (-0.019). Figure 8 also shows

a lighter SSA decrease with the BRC simulation at high latitudes such as over North America or Russia, yielding to an SSA in better agreement with the reference datasets (see white hatching). One has to note that the SSA of the various reference datasets differ significantly (see also Figure A1). Both on annual average or for the JAS season, the OMI product shows higher SSA (about +0.04) than those observed by PARASOL-GRASP and MACv2, over AFR and AME. Previous studies have already shown differences, sometimes systematic, between AERONET and satellite datasets and also among satellite products





(Schutgens et al., 2020, 2021). The NOBRC SSA during JAS, with averaged values of $0.930 \pm 0.001$ over AFR and $0.960 \pm 0.002$ over AME, is overestimated compared to all reference datasets. The BRC simulation, characterized by an SSA of $0.906 \pm 0.002$ ($0.941 \pm 0.004$) over AFR (AME) during this season, is therefore more consistent with the reference datasets, with SSA close to those measured by OMI ($0.914 \pm 0.002$ over AFR and $0.936 \pm 0.003$ over AME) and slightly higher than SSA derived by PARASOL-GRASP ($0.885 \pm 0.009$ over AFR and $0.903 \pm 0.009$ over AME) and MACv2 ($0.873 \pm 0.001$ over AFR and $0.884 \pm 0.001$ over AME).

Figure 9 presents the simulated AOD (550 nm) (NOBRC and BRC simulations) and that from the FMI_SAT, MACv2, PARASOL-GRASP and OMI datasets, averaged over the year and over the JAS season. Here too, the reference datasets differ significantly, as illustrated in Figure A2 that shows the AOD differences with FMI_SAT. Figure 9 shows close annual AODs between NOBRC ($0.150 \pm 0.003$ over AFR and $0.087 \pm 0.009$ over AME) and BRC ($0.147 \pm 0.003$ over AFR and $0.092 \pm 0.009$ over AME). Similar results exist for JAS (see Table 5). The two model runs generally underestimate AOD values compared to the satellite products. For example, NOBRC and BRC present respectively a mean annual AOD of $0.150 \pm 0.003$ and $0.147 \pm 0.003$ over AFR while the reference datasets are comprised between $0.156 \pm 0.004$ and $0.360 \pm 0.010$.

Our last comparisons concerns the total AAOD (440 nm), and we compare the simulated AAOD with that of the PARASOL-GRASP and OMI satellite products and with the MACv2 dataset (see Figure 10). As for the SSA and the AOD, we highlight differences between these three observational datasets. Indeed, the PARASOL-GRASP dataset indicates an annual averaged AAOD of $0.045 \pm 0.003$ ($0.041 \pm 0.008$) over the AFR (AME) region, while the corresponding values of OMI and MACv2 are of $0.041 \pm 0.001$ and $0.031 \pm 0.001$ ($0.023 \pm 0.001$ and $0.026 \pm 0.001$). Differences between these datasets can also be observed in other regions such as Australia or China. Compared to these datasets, the NOBRC simulation shows too low annual AAOD over AFR ($0.010 \pm 0.001$) and AME ($0.004 \pm 0.001$) while the BRC simulation indicates higher values of $0.014 \pm 0.001$ ($0.007 \pm 0.001$) over AFR (AME), more consistent with the reference datasets (AAOD differences between BRC and reference datasets are shown in Figure A3).

To summarize, the comparisons of the different ARPEGE-Climat simulations indicate a SSA decrease as well as an AAOD increase thanks to the BrC implementation. This leads to a better agreement with all reference datasets both locally and on a larger scale, especially when the bleaching parameterization is taken into account. These changes are nevertheless maximum over the Tropics where large quantities of biomass burning aerosols are emitted.

## 4.2 BrC radiative and climatic effects

### 4.2.1 Brown carbon radiative effect

Figure 11 presents the mean annual and JAS effective radiative forcing from aerosol-radiation interactions (ERFari, see Eq. (4)), in clear-sky and all-sky conditions, and from aerosol–cloud interactions (ERFaci, Eq. (5)) based on the difference between the BRC and NOBRC simulations. Total ERF, in all-sky conditions, is also presented in this figure. The statistical test applied here is the Wilks test (Wilks, 2006, 2016). All radiative forcing estimates are summarized, for the different regions studied, in Table 6. In clear-sky conditions,the ERFari annual global mean is of $0.028 \pm 0.013$ W m$^{-2}$ while we compute a stronger





value ($0.064 \pm 0.022$ W m$^{-2}$) during the JAS period. During this season, the highest ERFari are found over regions impacted by BrC, which result in statistically significant warming effects of $0.404 \pm 0.100$ W m$^{-2}$ and of $0.358 \pm 0.111$ W m$^{-2}$ on average over the AFR region, which has the highest BB emissions (see Figure 2), and over the AME region respectively. In clear-sky conditions, the highest values are found over the continents where they can reach up to 1.5 W m$^{-2}$. Clear-sky annual ($0.029 \pm 0.006$ W m$^{-2}$) and JAS ($0.062 \pm 0.011$ W m$^{-2}$) global averages are similar to all-sky ones. Over the AFR region, Figure 11 shows statistically significant larger ERFari ($0.292 \pm 0.034$ W m$^{-2}$ in annual mean and $0.785 \pm 0.110$ W m$^{-2}$ over JAS), notably due to high values over the Atlantic Ocean, up to 1 (4) W m$^{-2}$ on annual (JAS) mean, which are due to the presence of stratocumulus, and therefore of high albedo. Indeed, in addition to being absorbed in smoke plumes, incoming solar radiation is also reflected by stratocumulus clouds and absorbed again by absorbing biomass burning aerosol (Abel et al., 2005; Zhang et al., 2016). Our results are consistent with those of Brown et al. (2018) who analyse simulations that include or not a bleaching parameterization. Brown et al. (2018) also indicate a positive ERFari on annual average, especially over the AFR region with maxima comprised between 0.8 and 1.7 W m$^{-2}$ depending on the implementation or not of the BrC bleaching effect. At the global scale, they show annual mean ERFari between 0.06 (with BrC bleaching effect) and 0.13 W m$^{-2}$ (without BrC bleaching effect). Using the same bleaching parameterization as Brown et al. (2018) in the GEOS-Chem CTM, Wang et al. (2018) calculated an annual global BrC IRF of 0.05 W m$^{-2}$. Lastly, other studies report annual global BrC direct radiative effects between 0.10 and 0.12 W m$^{-2}$ (Feng et al., 2013; Wang et al., 2014; Saleh et al., 2015; Jo et al., 2016; Zhang et al., 2020).

In contrast to ERFari patterns which are statistically significant and well co-localized with the BB BrC sources, the effective radiative forcing from aerosol–cloud interactions (ERFaci), also shown in Figure 11, appears less clearly. The mean annual (JAS) global ERFaci is of - $0.024 \pm 0.066$ W m$^{-2}$ ($0.016 \pm 0.154$ W m$^{-2}$), therefore too noisy to be meaningful. All ERFaci values, over the different regions studied, are also summarized in Table 6. It is important to remember here that the BrC indirect effect is taken into account in the same way as that of the OA aerosols (hydrophilic bin). In comparison to our results, Brown et al. (2018) found an annual global ERFaci of $0.01 \pm 0.04$ W m$^{-2}$ (both with and without a BrC bleaching effect). Over the AFR region, our ERFaci is of - $0.328 \pm 0.227$ W m$^{-2}$ on annual average (- $0.571 \pm 0.493$ W m$^{-2}$ over the JAS period) with the most negative values on the ocean side, suggesting that the BrC absorption increases the low cloud formation and lifetime in the model there. These results are consistent with the Sakaeda et al. (2011) and Johnson et al. (2004) studies which show that, over ocean, the presence of absorbing aerosols above clouds, that increases the potential temperature above the cloud top and therefore creates less favourable conditions for cloud top entrainment, allows for a more persistent low marine cloud cover. Over the AME region, we derive a positive ERFaci of $0.207 \pm 0.331$ W m$^{-2}$ on annual average ($0.150 \pm 0.642$ W m$^{-2}$ over the JAS period), therefore suggesting a decrease in cloud formation that is also consistent with the Sakaeda et al. (2011) study. Further insight on the BrC effects on clouds in ARPEGE-Climat is presented in section 4.2.2. Finally, in terms of total ERF, we obtain positive annual values of $0.028 \pm 0.116$ W m$^{-2}$ at the global scale, of $0.190 \pm 0.294$ W m$^{-2}$ over the AFR region and of $0.156 \pm 0.242$ W m$^{-2}$ over the AME region.





### 4.2.2 Brown carbon effects on temperature and low clouds

Figure 12 and Table 7 present the BrC JAS and annual impacts on low-level (below 640 hPa) cloud fraction and temperature at 700 hPa, calculated as differences between the BRC and NOBRC simulations. For information, no significant BrC impact was found on other meteorological fields often studied such as large-scale/convective precipitation or high-level (above 440 hPa) cloud fraction. On annual average, Figure 12 shows no significant effect of BrC on the low-level cloud fraction (- 0.008 ± 0.102 %). In their study, Brown et al. (2018) rather show small positive low-level cloud fraction changes (not statistically significant though), between 0.03 and 0.06 ± 0.03 % depending on their BrC parameterization. More regionally and on seasonal average, Figure 12 shows an increase of the low-level cloud fraction over the AFR region (0.645 ± 0.342 %) and even more over the OCE region (up to 2 % with an averaged value of 0.723 ± 0.799 %) during JAS. This OCE region (OCEan, 15°W - 10°E / 0 - 25°S) is particularly interesting here because it corresponds to the maxima in BrC radiative forcing. It is therefore the region where the rapid responses of the atmosphere will be due to the BrC aerosol plume. The low-level cloud fraction increase over the OCE region during JAS is well correlated to a vertical velocity decrease of about 10% (- 0.004 Pa s$^{-1}$) at 700 hPa (see Figure 13). This vertical velocity decrease is consistent with a low-level tropospheric stability increase leading to a larger low cloud cover in the marine boundary layer. Our results are consistent with those of several studies showing large changes in vertical velocity caused by the absorption of smoke aerosols in the lower troposphere, that could influence the low marine cloud cover. Indeed, a low-level cloud cover increase (decrease) is related to negative (positive) vertical velocity changes (Sakaeda et al., 2011; Wilcox, 2012; Brown et al., 2018; Allen et al., 2019; Deaconu et al., 2019; Mallet et al., 2020). Our study shows also that the augmentation in low-level cloud fraction during JAS over the OCE region is co-located with a negative ERFaci of - 0.739 ± 0.934 W m$^{-2}$ (see Figure 11). This ERFaci low-level cloud fraction relation is also present over other regions of the globe, such as South America, Europe, North America, India or China. We computed a global spatial correlation of - 0.42 (annual average) and - 0.47 (JAS average) between these two parameters. The ERFaci low-level cloud fraction relation was also shown in Brown et al. (2018) study (no value given) over various regions of the world such as South America, western Australia, the Middle East or the northeastern China.

Lastly, temperature changes at 700 hPa (altitude with most smoke aerosol transport, Das et al. 2017) due to the BrC addition into the model are shown in Figure 12. Small and no statistically significant effects can be seen on annual averages (- 0.001 ± 0.026°C global mean). Over JAS, Figure 12 shows statistically significant larger changes, up to + 0.6°C in northern Russia and up to 0.5°C in the south of the OCE region. Conversely, statistically significant temperature decreases, up to - 0.5°C, can be observed, as for example in eastern China. Over these regions, the temperature shifts would rather be related to climate feedbacks, teleconnections or even changes in atmospheric dynamics but not directly to BrC aerosol plumes. Over the OCE region, directly impacted by a BrC aerosol plume, the temperature increase (0.099 ± 0.101°C) is mostly caused by smoke aerosol solar absorption associated with a (solar) heating rate increase of about 0.1 K per day (11%, see Figure 13). Above 800 hPa, this heating rate increase is largely due to the absorption of BrC particles while between 800 and 850 hPa, the heating rate increase is also due in part to the low-level cloud fraction increase described previously. It still remains important to remind here





that the ARPEGE-Climat global atmospheric model is run in an AMIP-type mode, and therefore coupled ocean-atmosphere simulations would therefore be relevant to consolidate, or not, our results.

## 5   Conclusions

Organic aerosols have long been considered to be only scattering aerosols, but recent studies have shown that a part of these
aerosols, called brown carbon particles, could absorb solar radiation. In parallel, several multi-model studies have led to the conclusion that models underestimate the AAOD: Shindell et al. (2013) show an AAOD underestimation of the ACCMIP models compared to AERONET stations, especially over South America and Southern Hemisphere Africa, while very recently, Mallet et al. (2021) demonstrate that the majority of CMIP6 global climate models underestimates the absorption of biomass-burning aerosols over the Southeast Atlantic. With emerging evidence of the importance of BrC, especially for solar absorption
at UV and short visible wavelengths, all this has led to the inclusion of BrC in chemical transport models and more recently in some global climate models.

Here we have implemented a BrC parameterization, derived from Saleh et al. (2014), into the TACTIC aerosol scheme of ARPEGE-Climat, the atmospheric component of the CNRM global climate model. We have conducted a BrC radiative and climatic effect study thanks to several simulations over the period 2000-2014. The implementation of a BrC parameterization
in climate models can be done in several ways. The method we adopted, which allows a good representation of the spatial and temporal variability of the BrC absorption, consists in parameterizing the BrC imaginary refractive index according to the BC-to-OA ratio in emissions (Saleh et al., 2015; Brown et al., 2018; Wang et al., 2018). As recent studies have shown broadly similar results using, or not, a constant BC-to-OA ratio (Brown et al., 2018; Wang et al., 2018), we have therefore chosen to use a constant (0.08) BC-to-OA ratio. The effect of bleaching has also been included.

The implementation of BrC particles into TACTIC has allowed to significantly improve several total aerosol optical properties, such as the total aerosol SSA and AAOD, both at the local and regional scales, particularly in regions with high biomass burning emissions (AFR and AME) which are the main sources of BrC. However, at certain African and South-American AERONET sites, climatological AAOD are still slightly underestimated by our model. One hypothesis to explain our bias is the spatial and temporal representativeness of the AERONET data. It is also likely that part of the differences between the
model and the reference datasets could come from the emission inventories used in this study. We have outlined however a strong disparity between the different reference datasets used in this work, ground-based and satellite data (PARASOL-GRASP, OMI, MACv2, FMI_SAT) on all the studied parameters, and this indicates the need of further work to provide modellers with solid reference datasets.

We computed annual and JAS BrC ERFs (ERFari, ERFaci and total ERF) both at the global and regional scales. In all-sky
conditions, the annual global BrC ERFari is of $0.029 \pm 0.006$ W m$^{-2}$ while the BrC ERFaci is of $-0.024 \pm 0.066$ W m$^{-2}$. At the regional scale, over regions impacted by BrC, our BrC ERFari warming effect goes up to $0.085 \pm 0.032$ W m$^{-2}$ (AME region) and to $0.292 \pm 0.034$ W m$^{-2}$ (statistically significant, AFR region). While ERFari patterns are statistically significant and well localized with BrC sources, ERFaci patterns are more patchy with a lot of regional differences, ranging from - 2.5 to





2.5 W m$^{-2}$. We derive from our study a small annual global BrC ERF of 0.028 $\pm$ 0.116 W m$^{-2}$, with maximum values over the AFR region (0.190 $\pm$ 0.294 W m$^{-2}$). Our ERF results compare relatively well with those of Brown et al. (2018) when Brown et al. (2018) considers a BrC bleaching parameterization that reduces by half their BrC radiative forcing. While we have some confidence in comparing our study with that of Brown et al. (2018) that follows to some extent the protocol we use (Ghan (2013) formulation, AMIP-type simulations with and without BrC), it does not seem appropriate to compare our results with others in literature (see our introduction) that use different methodologies or present different radiative forcing concepts (e.g., IRF). We believe that our results which follow the CMIP6 AerChemMIP (Collins et al., 2017) ) and RFMIP (Pincus et al., 2016) recommendations are solid.

Taking into account the BrC effect, and therefore the absorption increase of biomass burning plumes, contributes moderately to a low-level cloud fraction change. Indeed, a slight low-level cloud fraction increase (0.723 $\pm$ 0.799 %) was simulated over the OCE region during JAS. This low-level cloud fraction increase is due to a low-level tropospheric stability increase, caused at least in part by a 700 hPa vertical velocity decrease of about 10%. Mallet et al. (2021) have shown that the frequent underestimation of biomass-burning aerosol absorption by climate models over the Southeast Atlantic could lead to a misrepresentation of the low-level cloud response. Their study highlights the importance for climate models to properly represent the biomass-burning aerosol absorption, and consequently the need to consider BrC aerosols. Finally in terms of climatic impacts, we simulated a moderate 700 hPa temperature increase (0.099 $\pm$ 0.101°C) over the OCE region during JAS. This temperature augmentation is caused non only by a heating rate increase of about 11% but also by the low-level cloud fraction increase over this region.

Results shown in this study highlight the importance of taking the BrC aerosols into account in climate models. Nevertheless, several hypotheses would need to be tested to further evaluate our model sensitivity and possibly improve the model in terms of both radiative and climatic impacts. These hypotheses include the BrC lifetime, or the use of different BC-to-OA ratios for BB and BF emissions that would allow to differentiate the BrC optical properties according to the source of emissions. We have to note however that further studies are needed, in addition to those already published (e.g., Zhong and Jang (2011); Lee et al. (2014); Forrister et al. (2015); Zhao et al. (2015); Wang et al. (2016); Vakkari et al. (2018)), to improve the knowledge about BrC, for instance BrC aging processes (e.g., formation of secondary BrC, lifetime, evolution of the absorption). Finally, fully coupled atmosphere-ocean climate model simulations would be relevant to allow BrC absorption to affect air–sea interactions.

*Code and data availability.* This study relies entirely on publicly available data. Codes of the various components of the ARPEGE-Climat model are available as follows: the SURFEX code is accessible using a CECILL-C Licence (http://www.cecill.info/licences/Licence_CeCILL-C_V1-en.txt) at http://www.umr-cnrm.fr/surfex; OASIS3-MCT is available at https://verc.enes.org/oasis/download; XIOS at https://forge.ipsl-.jussieu.fr/ioserver and the rest of the ARPEGE-Climat code is available upon request to the authors. The output of the different simulations presented here are available upon request from the authors (thomas.druge@meteo.fr). PARASOL-GRASP data are publicly available on the official GRASP algorithm website at https://www.grasp-open.com/products (last access: 23 February 2022). OMI product can be obtained from the NASA Earth data portal at https://earthdata.nasa.gov/ (last access: 23 February 2022). MACv2 product is available at ftp://ftp-projects.mpimet.mpg.de/aerocom/climatology/MACv2_2018/ (last access: 23 February 2022). FMI_SAT data are available at





http://nsdc.fmi.fi/data/data_aod (last access: 23 February 2022). Lastly, AERONET data are available at https://aeronet.gsfc.nasa.gov/ (last access: 23 February 2022).

*Author contributions.* All authors designed the experiment methodology and TD carried them out. TD wrote the paper with contributions from all co-authors.

5 *Competing interests.* The authors declare that they have no conflict of interest.

*Acknowledgements.* This work has received funding from the EU funded Copernicus Atmospheric Monitoring Services 43 (CAMS_43, "global aerosol aspects") project as well as from the European Union's Horizon 2020 research and innovation programme under Grant Agreement N° 101003536 (ESM2025–Earth System Models for the Future). We thank the principal investigators of the AERONET network and his staff for establishing and maintaining the different sites used in this investigation. All providers of satellite data are acknowledged
10 for making available their aerosol datasets. We also acknowledge the support of the entire team in charge of the CNRM climate models. Supercomputing time was provided by the Météo-France/DSI supercomputing center.





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





| Station | Location | Altitude (m) | Number of months 3 years (see text for details) | Total year available | Total JAS available |
|---|---|---|---|---|---|
| **Africa** | | | | | |
| 1 - Lubango (Angola) | 15.0S, 13.4E | 2047 | 7 | 4 | 4 |
| 2 - Ascension_Island (Ascension Island) | 8.0S, 14.4W | 30 | 7 | 8 | 7 |
| 3 - HESS (Namibia) | 23.3S, 16.5E | 1818 | 11 | 5 | 4 |
| 4 - Windpoort (Namibia) | 19.4S, 15.5E | 1206 | 9 | 4 | 4 |
| 5 - Skukuza (South Africa) | 25.0S, 31.6E | 265 | 8 | 11 | 11 |
| 6 - Mongu Inn (Zambia) | 15.3S, 23.1E | 1040 | 5 | 7 | 6 |
| **South America** | | | | | |
| 7 - Alta_Floresta (Brazil) | 9.9S, 56.1W | 277 | 6 | 19 | 17 |
| 8 - CUIABA-MIRANDA (Brazil) | 15.7S, 56.1W | 210 | 6 | 13 | 11 |
| 9 - Rio Branco (Brazil) | 10.0S, 67.9W | 212 | 4 | 11 | 11 |
| **Other** | | | | | |
| 10 - Jabiru (Australia) | 12.7S, 132.9E | 30 | 8 | 16 | 15 |
| 11 - Beijing (China) | 40.0N, 116.4E | 92 | 11 | 19 | 9 |
| 12 - XiangHe (China) | 39.7N, 117.0E | 36 | 12 | 17 | 16 |
| 13 - Dunkerque (France) | 51.0N, 2.4E | 5 | 8 | 15 | 9 |
| 14 - Kanpur (India) | 26.5N, 80.2E | 123 | 10 | 20 | 4 |
| 15 - Venise (Italy) | 45.3N, 12.5E | 10 | 12 | 21 | 17 |
| 16 - Moscow (Russia) | 55.7N, 37.5E | 192 | 7 | 17 | 13 |
| 17 - Granada (Spain) | 37.2N, 3.6W | 680 | 12 | 16 | 15 |
| 18 - Silpakorn_Univ (Thailand) | 13.8N, 100.0E | 72 | 6 | 15 | 0 |

**Table 1.** Characteristics of the AERONET stations used in this study: station name, location, altitude, the number of months available over at least 3 years during the observation period (2000-2020) and the total year/JAS available over the observation period (2000-2020).



|  | Resolution | Data period | SSA (nm) | AOD (nm) | AAOD (nm) |
|---|---|---|---|---|---|
| PARASOL-GRASP | 1° | 2006-2012 | 440 | 565 | 440 |
| OMI | 1° | 2005-2019 | 440 | 550 | 440 |
| MACv2 | 1° | 2001-2016 | 400 | 550 | 400 |
| FMI_SAT | 1° | 1995-2017 | X | 550 | X |
| AERONET | ground stations | 2000-2020 | 440 | 550 | 440 |

**Table 2.** Summary of the main characteristics of the reference datasets used in this study.



|  | Period of simulation | Members | BrC parameterization | Bleaching parameterization |
|---|---|---|---|---|
| NOBRC | 2000-2014 | 2 | No | No |
| BRC_NOBL | 2000-2014 | 2 | Yes | No |
| BRC | 2000-2014 | 2 | Yes | Yes |

**Table 3.** Summary of the main characteristics of the three simulations used in this study.





| | Africa | | | South America | | | Other | | |
|---|---|---|---|---|---|---|---|---|---|
| | SSA | AOD | AAOD | SSA | AOD | AAOD | SSA | AOD | AAOD |
| NOBRC (2000-2014) | 0.918 ± 0.001 | 0.271 ± 0.012 | 0.028 ± 0.001 | 0.948 ± 0.004 | 0.348 ± 0.058 | 0.027 ± 0.006 | 0.959 ± 0.001 | 0.264 ± 0.006 | 0.013 ± 0.001 |
| BRC_NOBL (2000-2014) | 0.865 ± 0.002 | 0.300 ± 0.014 | 0.050 ± 0.002 | 0.900 ± 0.008 | 0.375 ± 0.066 | 0.055 ± 0.014 | 0.954 ± 0.001 | 0.274 ± 0.005 | 0.014 ± 0.001 |
| BRC (2000-2014) | 0.890 ± 0.002 | 0.291 ± 0.013 | 0.040 ± 0.002 | 0.917 ± 0.008 | 0.361 ± 0.064 | 0.045 ± 0.012 | 0.955 ± 0.001 | 0.271 ± 0.008 | 0.014 ± 0.001 |
| AERONET (2000-2020) | 0.887 ± 0.010 | 0.238 ± 0.032 | 0.041 ± 0.006 | 0.907 ± 0.012 | 0.355 ± 0.124 | 0.041 ± 0.012 | 0.925 ± 0.009 | 0.279 ± 0.029 | 0.022 ± 0.003 |
| PARASOL–GRASP (2006-2012) | 0.876 ± 0.007 | 0.273 ± 0.027 | 0.050 ± 0.004 | 0.910 ± 0.012 | 0.497 ± 0.228 | 0.057 ± 0.017 | 0.896 ± 0.005 | 0.336 ± 0.035 | 0.041 ± 0.003 |
| OMI (2005-2019) | 0.904 ± 0.003 | 0.279 ± 0.015 | 0.039 ± 0.002 | 0.932 ± 0.004 | 0.392 ± 0.124 | 0.033 ± 0.008 | 0.932 ± 0.004 | 0.202 ± 0.008 | 0.020 ± 0.001 |
| MACv2 (2001-2016) | 0.876 ± 0.001 | 0.250 ± 0.013 | 0.059 ± 0.003 | 0.873 ± 0.001 | 0.310 ± 0.048 | 0.068 ± 0.010 | 0.930 ± 0.001 | 0.267 ± 0.009 | 0.034 ± 0.001 |
| FMI_SAT (1995-2017) | X | 0.205 ± 0.008 | X | X | 0.415 ± 0.073 | X | X | 0.333 ± 0.015 | X |

**Table 4.** SSA (440 nm), AOD (550 nm) and AAOD (440 nm) JAS averages simulated with the NOBRC, BRC_NOBL and BRC simulations and provided by reference datasets (PARASOL–GRASP, OMI, MACv2 and FMI_SAT, see Table 2 for details). Means ±2σ (significant level of 95%) values interpolated at the station location (see Table 1) are given.





| | | | NOBRC | BRC | PARASOL-GRASP | OMI | MACv2 | FMI_SAT |
|---|---|---|---|---|---|---|---|---|
| | | | (2000-2014) | (2000-2014) | (2006-2012) | (2005-2019) | (2001-2016) | (1995-2017) |
| SSA | Year | AFR | 0.952 ± 0.001 | 0.940 ± 0.001 | 0.888 ± 0.009 | 0.920 ± 0.002 | 0.890 ± 0.001 | X |
| | | AME | 0.976 ± 0.001 | 0.968 ± 0.002 | 0.888 ± 0.005 | 0.938 ± 0.003 | 0.890 ± 0.001 | X |
| | JAS | AFR | 0.930 ± 0.001 | 0.906 ± 0.002 | 0.885 ± 0.009 | 0.914 ± 0.002 | 0.873 ± 0.001 | X |
| | | AME | 0.960 ± 0.002 | 0.941 ± 0.004 | 0.903 ± 0.009 | 0.936 ± 0.003 | 0.884 ± 0.001 | X |
| AOD | Year | AFR | 0.150 ± 0.003 | 0.147 ± 0.003 | 0.257± 0.009 | 0.360 ± 0.010 | 0.156 ± 0.004 | 0.217 ± 0.007 |
| | | AME | 0.087 ± 0.009 | 0.092 ± 0.009 | 0.249± 0.061 | 0.250 ± 0.023 | 0.136 ± 0.007 | 0.199 ± 0.022 |
| | JAS | AFR | 0.268 ± 0.009 | 0.265 ± 0.010 | 0.401± 0.022 | 0.416 ± 0.011 | 0.261 ± 0.009 | 0.309 ± 0.012 |
| | | AME | 0.175 ± 0.031 | 0.181 ± 0.033 | 0.297± 0.105 | 0.266 ± 0.049 | 0.213 ± 0.022 | 0.244 ± 0.035 |
| AAOD | Year | AFR | 0.010 ± 0.001 | 0.014 ± 0.001 | 0.045 ± 0.003 | 0.041 ± 0.001 | 0.031 ± 0.001 | X |
| | | AME | 0.004 ± 0.001 | 0.007 ± 0.001 | 0.041 ± 0.008 | 0.023 ± 0.001 | 0.026 ± 0.001 | X |
| | JAS | AFR | 0.024 ± 0.001 | 0.035 ± 0.001 | 0.068 ± 0.004 | 0.050 ± 0.001 | 0.057 ± 0.002 | X |
| | | AME | 0.012 ± 0.003 | 0.019 ± 0.004 | 0.040 ± 0.010 | 0.025 ± 0.003 | 0.044 ± 0.005 | X |

**Table 5.** SSA (440 nm), AOD (550 nm) and AAOD (440 nm) annual or JAS averages $\pm 2\sigma$ (significant level of 95%) simulated with (BRC) and without (NOBRC) BrC and provided by reference datasets (PARASOL-GRASP, OMI, MACv2 and FMI_SAT, see Table 2 for details) over the AFR and AME regions (see Figure 1 for details).





|  |  | Year | JAS |
|---|---|---|---|
| Global | ERFari (clear-sky) | $0.028 \pm 0.013$ | $0.064 \pm 0.022$ |
|  | ERFari | $0.029 \pm 0.006$ | $0.062 \pm 0.011$ |
|  | ERFaci | $- 0.024 \pm 0.066$ | $0.016 \pm 0.154$ |
|  | ERF | $0.028 \pm 0.116$ | $0.093 \pm 0.206$ |
| AFR | ERFari (clear-sky) | $0.172 \pm 0.044$ | $0.404 \pm 0.100$ |
|  | ERFari | $0.292 \pm 0.034$ | $0.785 \pm 0.110$ |
|  | ERFaci | $- 0.328 \pm 0.227$ | $- 0.571 \pm 0.493$ |
|  | ERF | $0.190 \pm 0.294$ | $1.083 \pm 0.706$ |
| AME | ERFari (clear-sky) | $0.100 \pm 0.038$ | $0.358 \pm 0.111$ |
|  | ERFari | $0.085 \pm 0.032$ | $0.302 \pm 0.101$ |
|  | ERFaci | $0.207 \pm 0.331$ | $0.150 \pm 0.642$ |
|  | ERF | $0.156 \pm 0.242$ | $0.349 \pm 0.480$ |

**Table 6.** BrC ERFari (W m$^{-2}$, clear-sky and all-sky conditions), ERFaci (W m$^{-2}$, all-sky conditions) and ERF (W m$^{-2}$, all-sky conditions) annual and JAS averages $\pm 2\sigma$ (significant level of 95%) for the BRC simulation at the global scale and over the AME and AFR regions (see Figure 1 for details).



|  |  | Year | JAS |
|---|---|---|---|
| Global | Low-level cloud fraction | - 0.008 ± 0.102 | - 0.011 ± 0.137 |
| | Temperature | - 0.001 ± 0.026 | 0.004 ± 0.056 |
| AFR | Low-level cloud fraction | 0.303 ± 0.198 | 0.645 ± 0.342 |
| | Temperature | 0.015 ± 0.062 | 0.001 ± 0.102 |
| AME | Low-level cloud fraction | - 0.206 ± 0.455 | 0.103 ± 0.617 |
| | Temperature | 0.005 ± 0.40 | 0.034 ± 0.097 |
| OCE | Low-level cloud fraction | 0.477 ± 0.406 | 0.723 ± 0.799 |
| | Temperature | 0.029 ± 0.043 | 0.099 ± 0.101 |

**Table 7.** Annual and JAS averages $\pm 2\sigma$ (significant level of 95%) BrC effects (differences between BRC and NOBRC) on low-level cloud fraction (%, relative percentage difference of cloud fraction) and temperature (K, 700 hPa) at the global scale and over the AME, AFR and OCE regions (see Figure 1 for details).





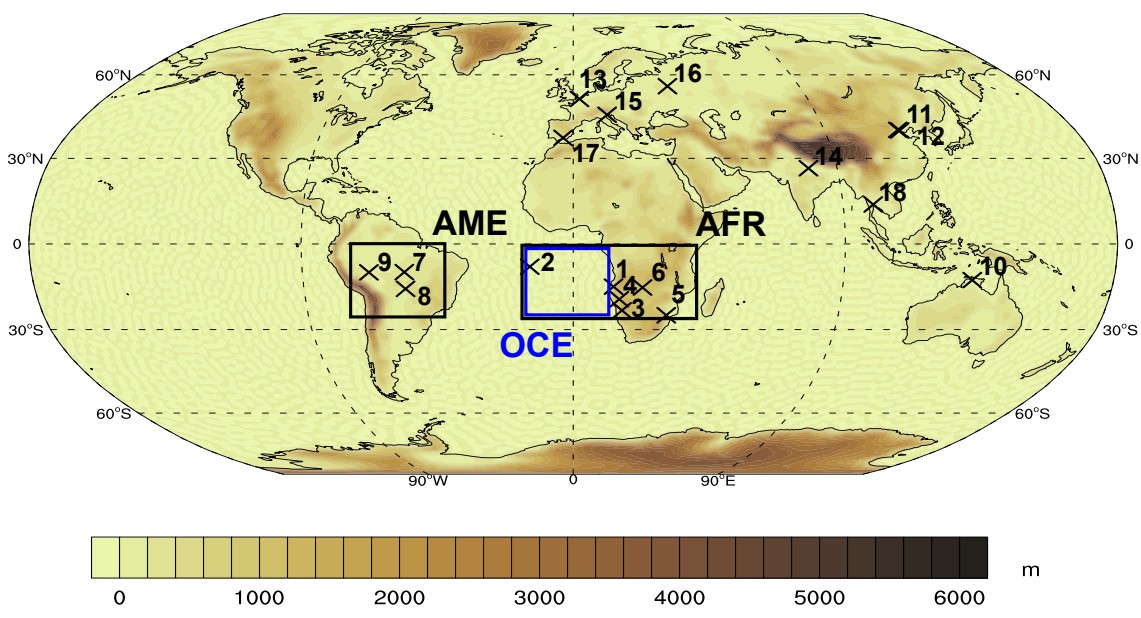

**Figure 1.** Orography (m) used in ARPEGE-Climat simulations. Locations of the Aerosol Robotic Network (AERONET) stations used in this study are shown (black crosses, see Table 1 for details on these AERONET stations), as the different regions used in this study, AME (70°-20°W / 0 - 25°S) and AFR (15°W - 40°E / 0 - 25°S, black boxes) and OCE (15°W - 10°E / 0 - 25°S, blue box).





**Figure 2.** Annual and JAS biomass burning (BB), biofuel (BF), and fossil fuel (FF) organic carbon emissions ($10^{-3}$ kg m$^{-2}$ mo$^{-1}$, 2000-2014 average). BB and BF emissions correspond to BrC emissions in the BRC_NOBL and BRC model runs. Total BB emissions (Tg) in the period are given on top of each panel.

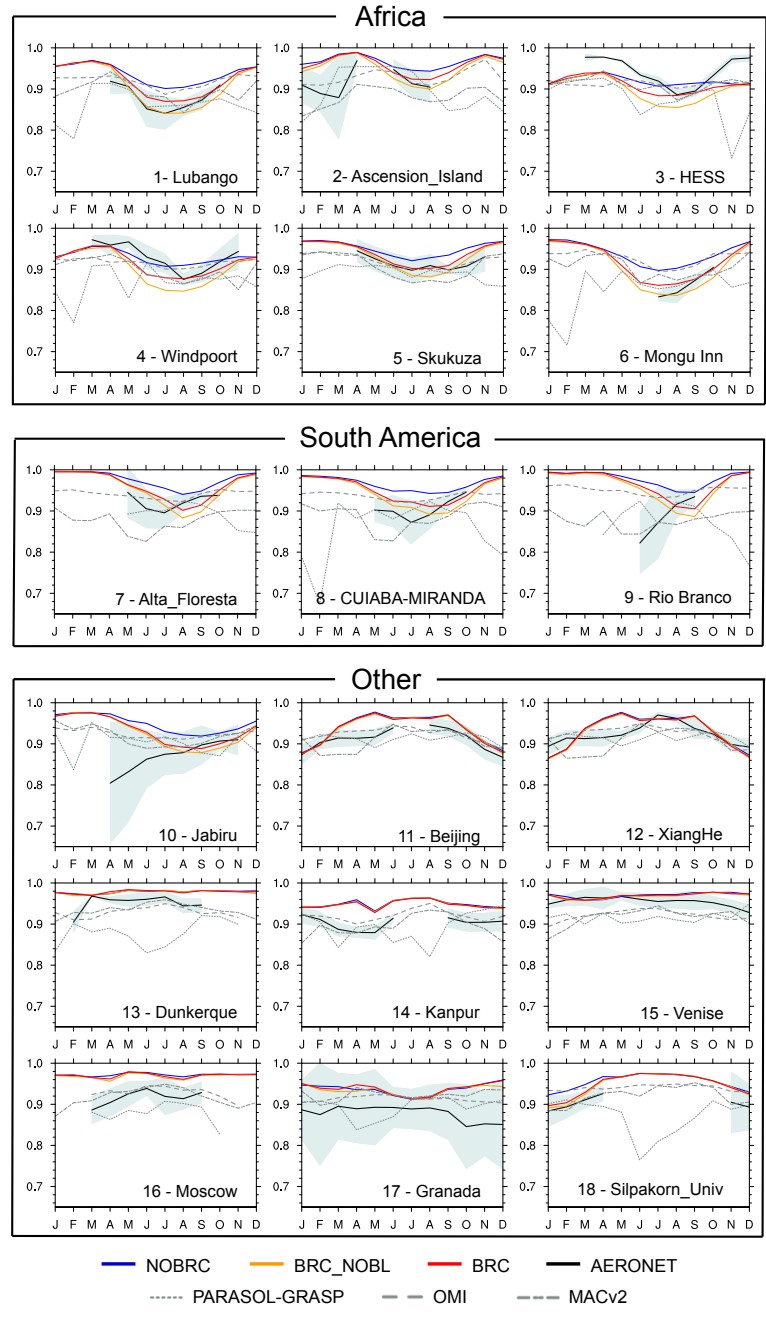

**Figure 3.** Annual cycle of the SSA (440 nm) at AERONET stations, simulated by the ARPEGE-Climat model for the NOBRC (blue), the BRC_NOBL (orange) and BRC (red) simulations (see Table 3 for details), with AERONET measurements (black, plus or minus standard deviation in light blue), and provided by reference datasets (grey) with PARASOL-GRASP, OMI and MACv2 (see Table 2 for details). Stations have been grouped into African ones, South American ones, and stations over the rest of the world (label "Other"). AERONET stations are detailed in Table 1.





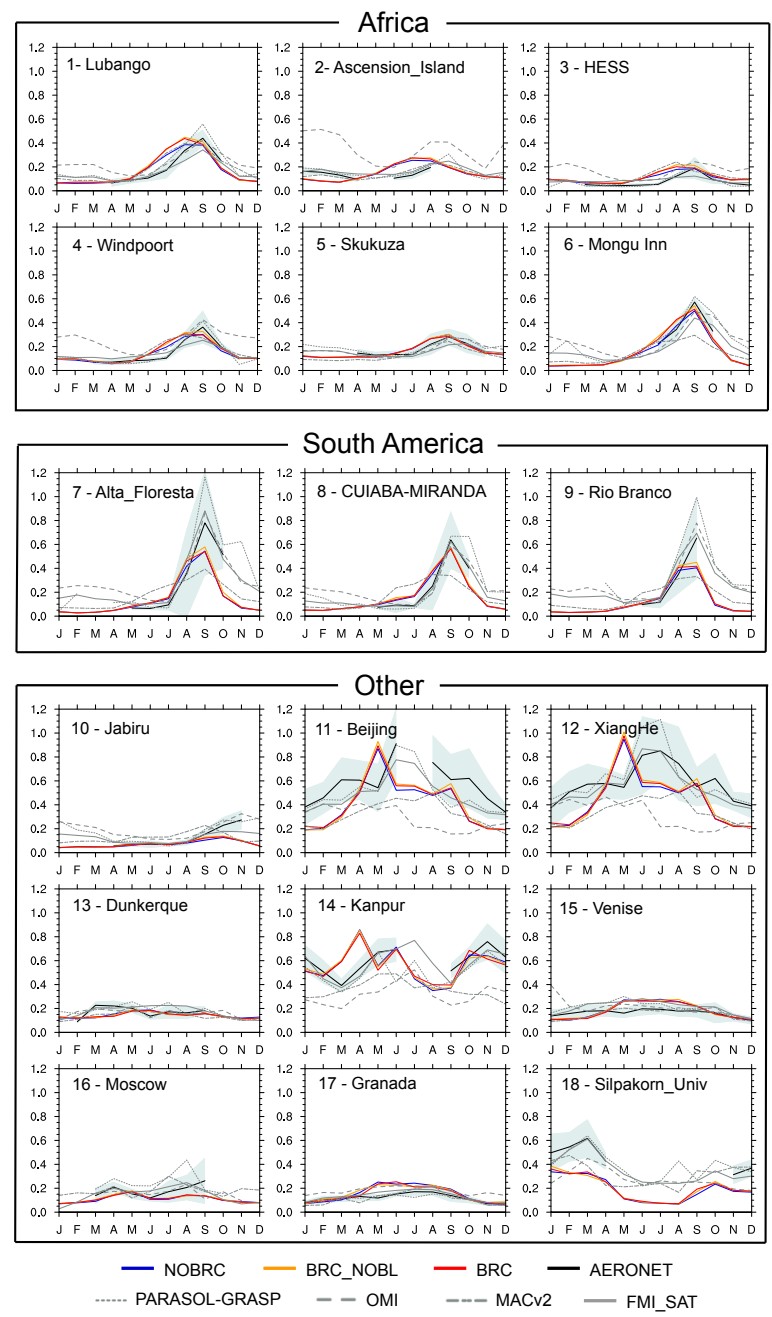

**Figure 4.** Same as Figure 3 but for the AOD (550 nm, see Tables 2 and 3 for details). FMI_SAT (grey) has been added here.





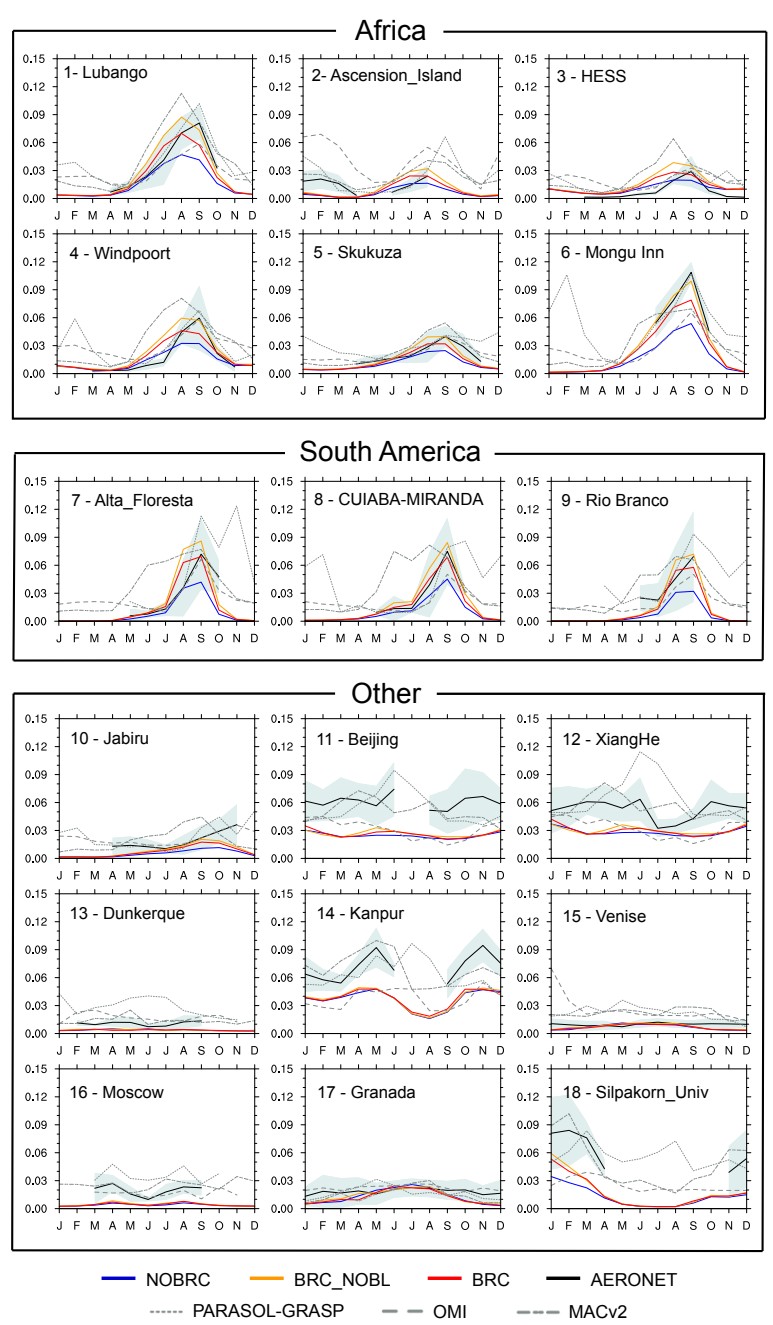

**Figure 5.** Same as Figure 3 but for the AAOD (440 nm, see Tables 2 and 3 for details).



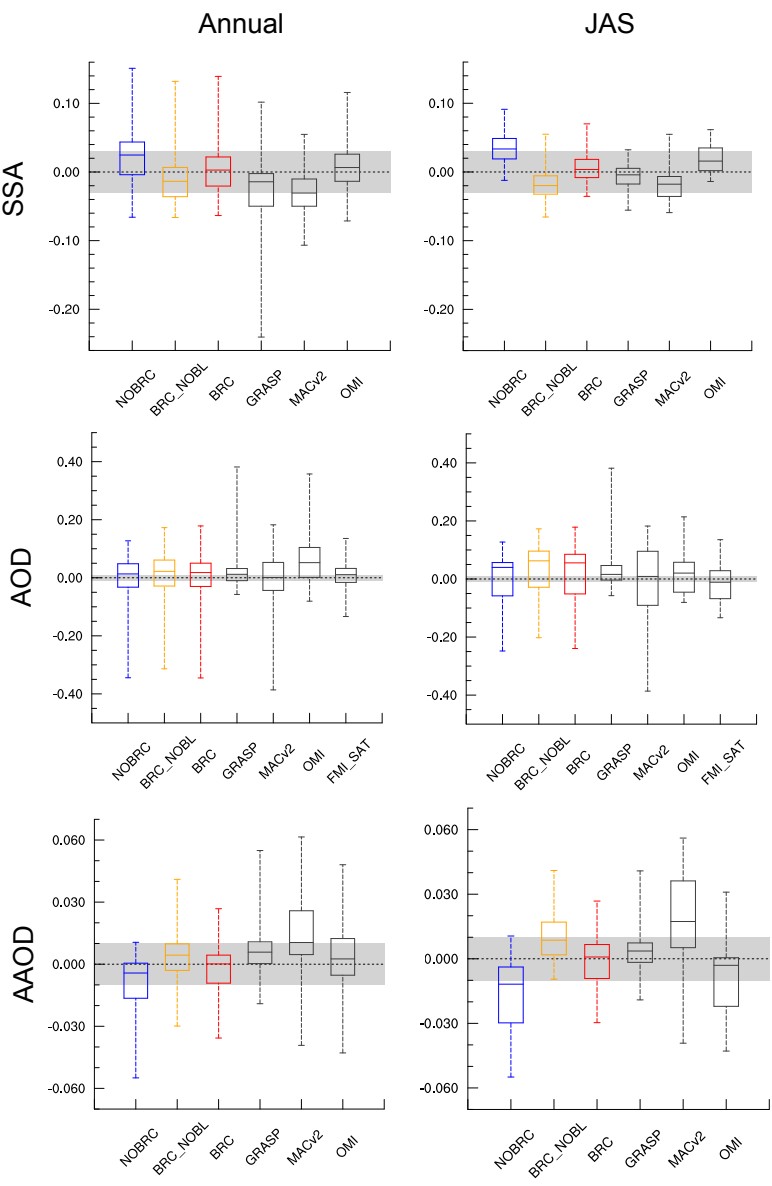

**Figure 6.** Boxplots of various biases compared to the AERONET observations, over the year (left column) and over JAS (right column). SSA (1st line), AOD (2nd line), AAOD (3rd line). Simulations: NOBRC, BRC_NOBL and BRC (see Table 3 for details), and reference data sets (PARASOL-GRASP, OMI, MACv2, FMI_SAT, see Table 2 for details). Uncertainty of the AERONET observations appears as shaded.





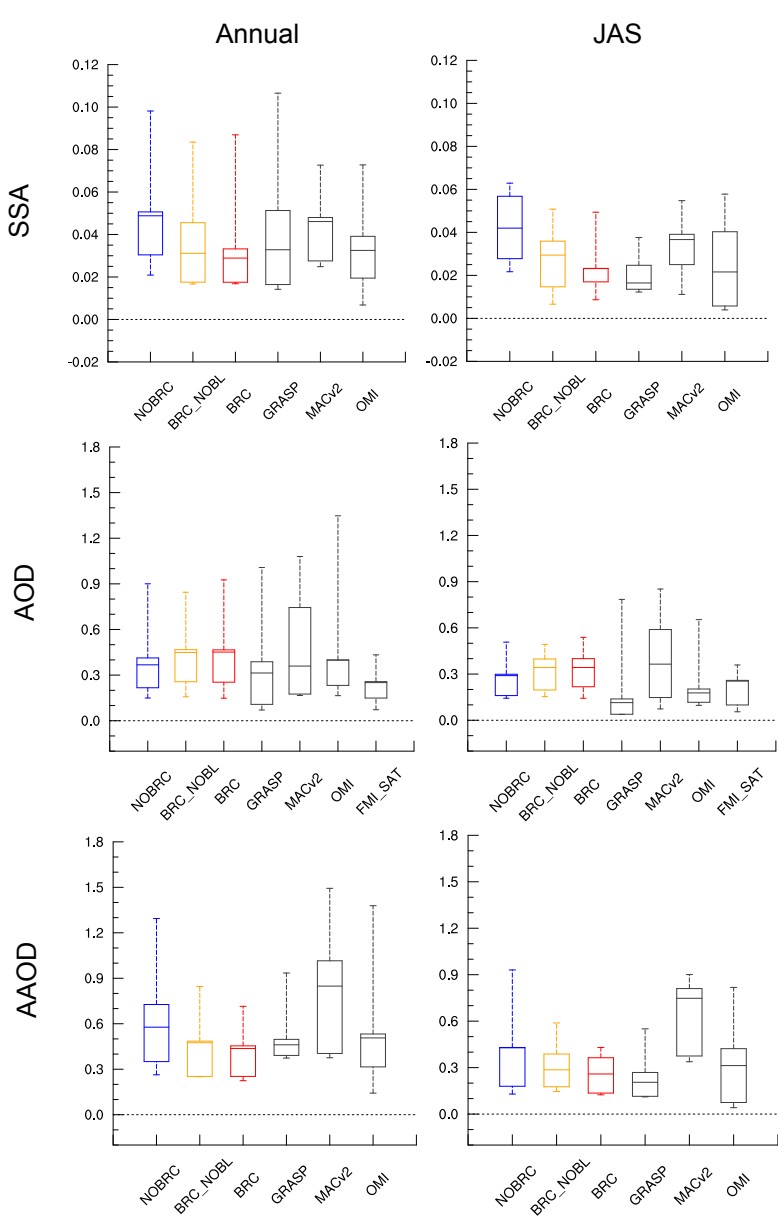

**Figure 7.** Same as Figure 6 but for the normalized root mean square error (NRMSE).

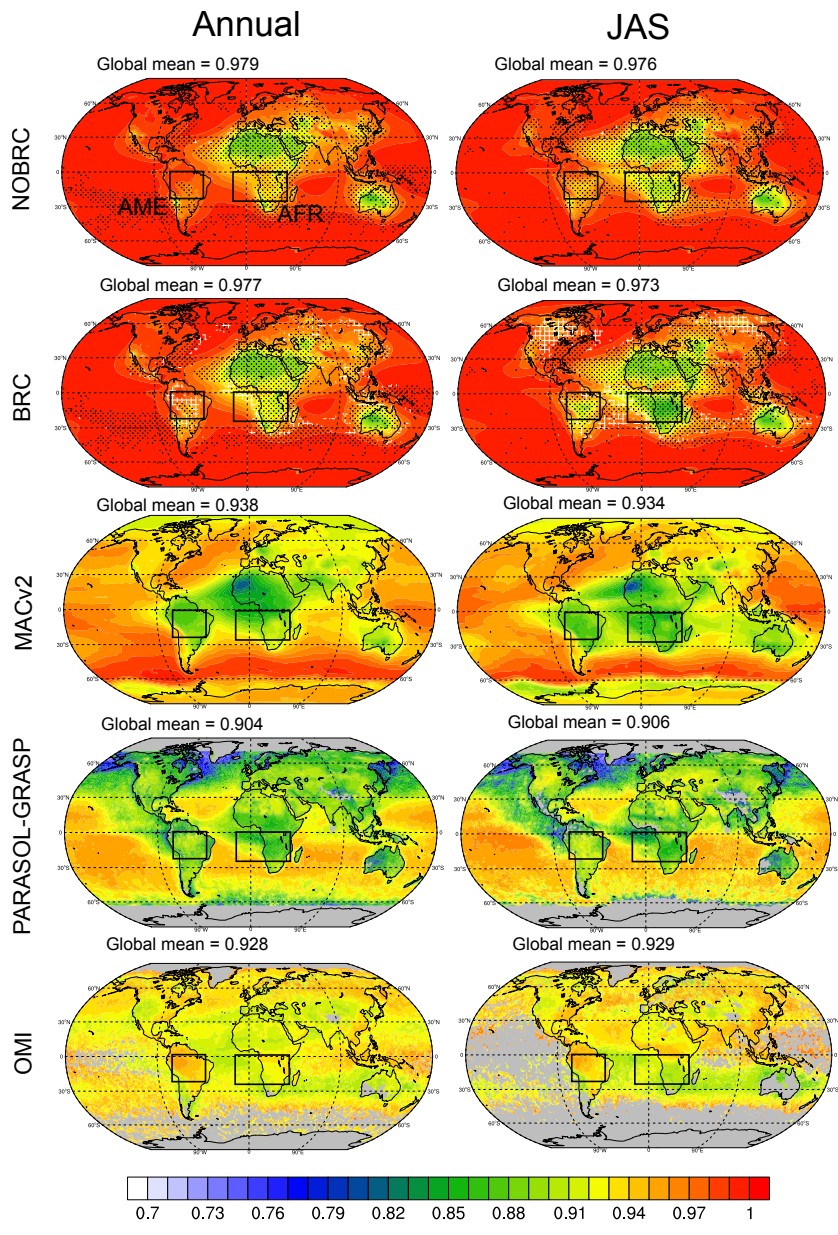

**Figure 8.** SSA (440 nm) simulated by the ARPEGE-Climat model for the NOBRC and BRC simulations and retrieved from MACv2, PARASOL-GRASP and OMI (see Tables 2 and 3 for details) averaged over the year and over the months of July, August and September (JAS). Black dots on the model plots show common locations between the NOBRC and BRC simulations where the SSA is inside the range covered by MACv2, PARASOL-GRASP and OMI (reference datasets uncertainties included, AERONET uncertainties are used when data are not available). Changes in these locations between the NOBRC and BRC simulations are highlighted by white hatching. Shaded areas indicate a missing value.







**Figure 9.** Same as Figure 8 but for the AOD (550 nm, see Tables 2 and 3 for details). FMI_SAT has also been added here.





**Figure 10.** Same as Figure 8 but for the AAOD (440 nm, see Tables 2 and 3 for details).





**Figure 11.** BrC ERFari (W m$^{-2}$, clear-sky and all-sky conditions) and ERFaci (W m$^{-2}$, all-sky conditions) and the sum of both (ERFari + ERFaci, W m$^{-2}$, all-sky conditions) averaged over the year and over the months of July, August and September (JAS). Hatching indicates regions with a significant effective radiative forcing at the 0.05 level (Wilks test).

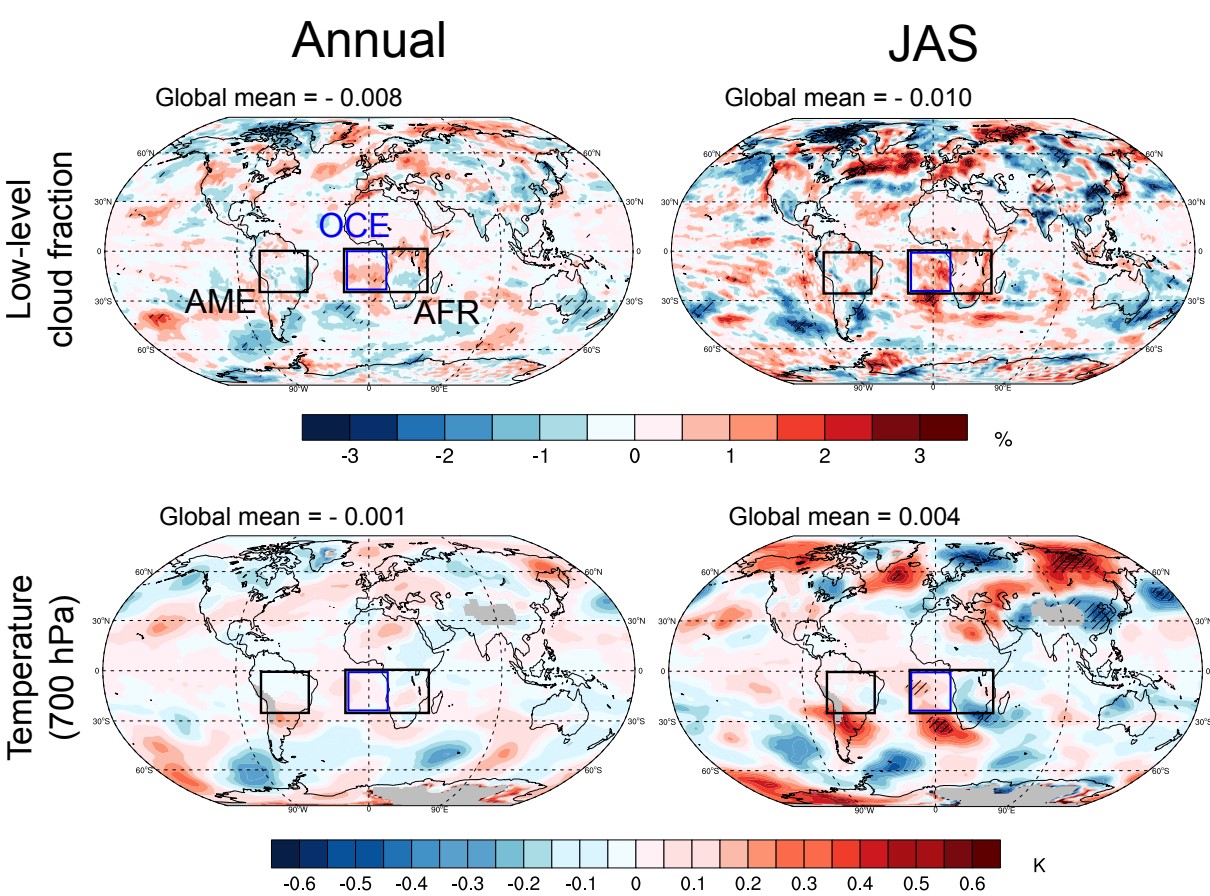

**Figure 12.** Changes between the BRC and NOBRC simulations (BRC minus NOBRC) in low-level (below 640 hPa) cloud fraction (%, relative percentage difference of cloud fraction) and temperature (K, 700 hPa) averaged over the year and over the months of July, August and September (JAS). Hatching indicates regions with a significant effect at the 0.05 level (Wilks test).



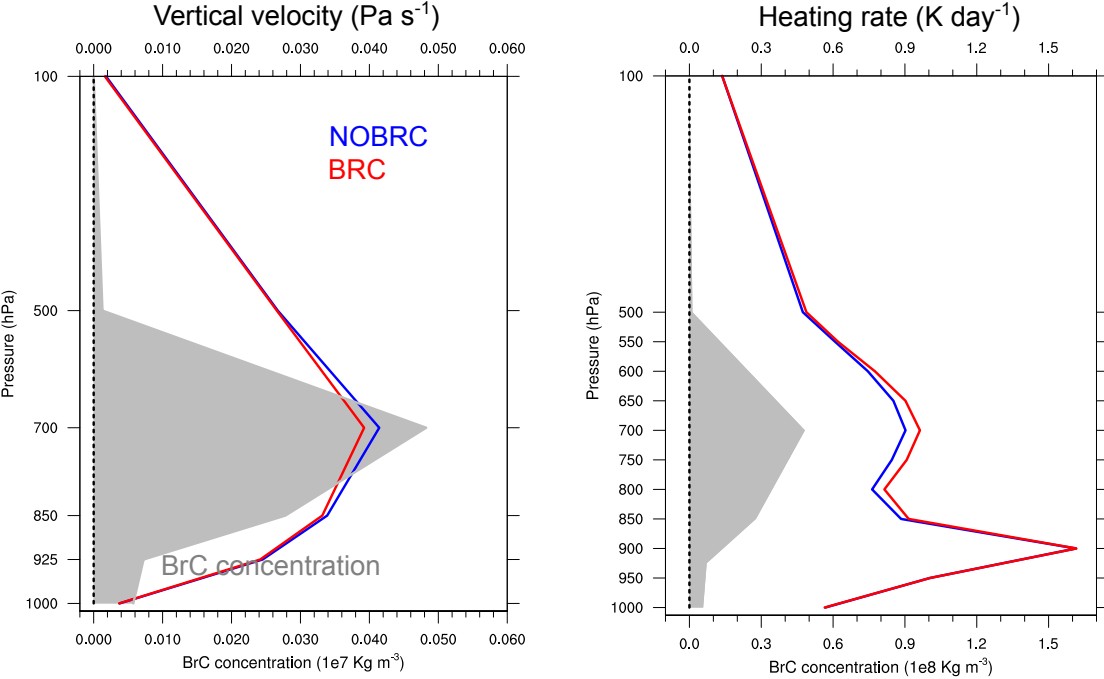

**Figure 13.** JAS mean over the OCE region of vertical velocity (Pa s$^{-1}$, left), and short-wave heating rate (K day$^{-1}$, right). BrC concentrations have been added in both figures (note the different X axis).



## Appendix A:

| Aerosol bin | 0% RH | | | | | | 80% RH | | | | | |
|---|---|---|---|---|---|---|---|---|---|---|---|---|
| | 350nm | | | 550nm | | | 350nm | | | 550nm | | |
| | Ext | SSA | g | Ext | SSA | g | Ext | SSA | g | Ext | SSA | g |
| BrC (hydrophobic) | 7.73 | 0.80 | 0.73 | 5.23 | 0.89 | 0.66 | 7.73 | 0.80 | 0.73 | 5.23 | 0.89 | 0.66 |
| BrC (hydrophilic) | 7.88 | 0.89 | 0.71 | 5.21 | 0.94 | 0.66 | 10.95 | 0.92 | 0.74 | 7.33 | 0.96 | 0.71 |
| BrC no bleaching (hydrophilic) | 7.73 | 0.80 | 0.73 | 5.23 | 0.89 | 0.66 | 10.79 | 0.85 | 0.76 | 7.34 | 0.92 | 0.71 |
| OA (hydrophobic) | 6.08 | 1.00 | 0.67 | 3.64 | 1.00 | 0.64 | 6.08 | 1.00 | 0.67 | 3.64 | 1.00 | 0.64 |
| OA (hydrophilic) | 5.08 | 1.00 | 0.65 | 3.02 | 1.00 | 0.62 | 11.09 | 1.00 | 0.73 | 6.87 | 1.00 | 0.71 |
| BC | 17.45 | 0.30 | 0.43 | 10.22 | 0.22 | 0.35 | 17.45 | 0.30 | 0.43 | 10.22 | 0.22 | 0.35 |
| SO$_4$ | 8.52 | 1.00 | 0.64 | 4.81 | 1.00 | 0.61 | 21.30 | 1.00 | 0.73 | 13.08 | 1.00 | 0.71 |
| NO$_3$ (fine) | 7.05 | 1.00 | 0.61 | 4.30 | 1.00 | 5.84 | 13.00 | 1.00 | 0.71 | 8.13 | 1.00 | 0.68 |
| NO$_3$ (coarse) | 0.19 | 0.85 | 0.84 | 0.19 | 0.89 | 0.82 | 0.54 | 1.00 | 0.85 | 0.55 | 1.00 | 0.85 |
| NH$_4$ | 6.20 | 1.00 | 0.64 | 3.50 | 1.00 | 0.61 | 15.49 | 1.00 | 0.73 | 9.51 | 1.00 | 0.71 |
| SS (0.01-1.0) | 3.64 | 0.98 | 0.73 | 2.57 | 1.00 | 0.71 | 9.86 | 0.99 | 0.77 | 8.05 | 1.00 | 0.79 |
| SS (1.0-10.0) | 0.58 | 0.85 | 0.82 | 0.59 | 1.00 | 0.75 | 1.40 | 0.93 | 0.83 | 1.43 | 1.00 | 0.81 |
| SS (10.0-100.0) | 0.41e-2 | 0.56 | 0.76 | 0.41e-2 | 0.98 | 0.72 | 0.01 | 0.63 | 0.77 | 0.01 | 0.99 | 0.76 |
| DD (0.01-1.0) | 4.54 | 0.97 | 0.64 | 1.93 | 0.98 | 0.55 | 4.54 | 0.97 | 0.64 | 1.93 | 0.98 | 0.55 |
| DD (1.0-2.5) | 1.02 | 0.84 | 0.79 | 1.00 | 0.95 | 0.67 | 1.02 | 0.84 | 0.79 | 1.00 | 0.95 | 0.67 |
| DD (2.5-20.0) | 0.15 | 0.67 | 0.88 | 0.15 | 0.83 | 0.83 | 0.15 | 0.67 | 0.88 | 0.15 | 0.83 | 0.83 |

**Table A1.** Aerosol optical properties, namely extinction cross section (Ext in m$^2$ g$^{-1}$), single scattering albedo (SSA) and asymmetry parameter (g), used in the present version of TACTIC, at 350 and 550 nm. Values are given for relative humidities of 0 and 80 %. Bin diameter limits ($\mu$m) are given in parentheses for SS and DD aerosols.

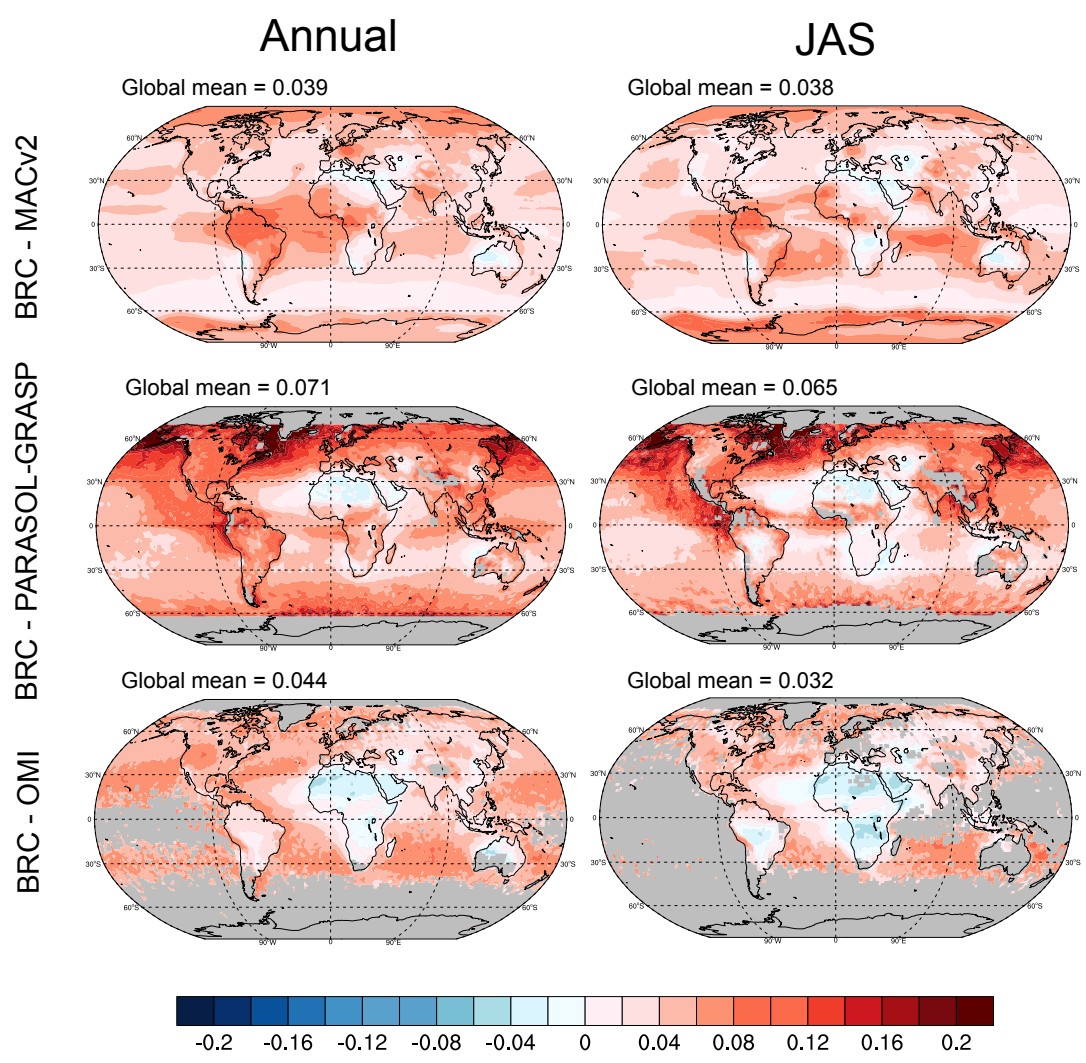

**Figure A1.** SSA (440 nm, see Tables 2 and 3 for details) difference between the BRC simulation and MACv2, PARASOL-GRASP and OMI averaged over the year and over the months of July, August and September (JAS).

**Figure A2.** AOD (550 nm, see Tables 2 and 3 for details) difference between FMI_SAT and the BRC simulation, MACv2, PARASOL-GRASP and OMI averaged over the year and over the months of July, August and September (JAS).



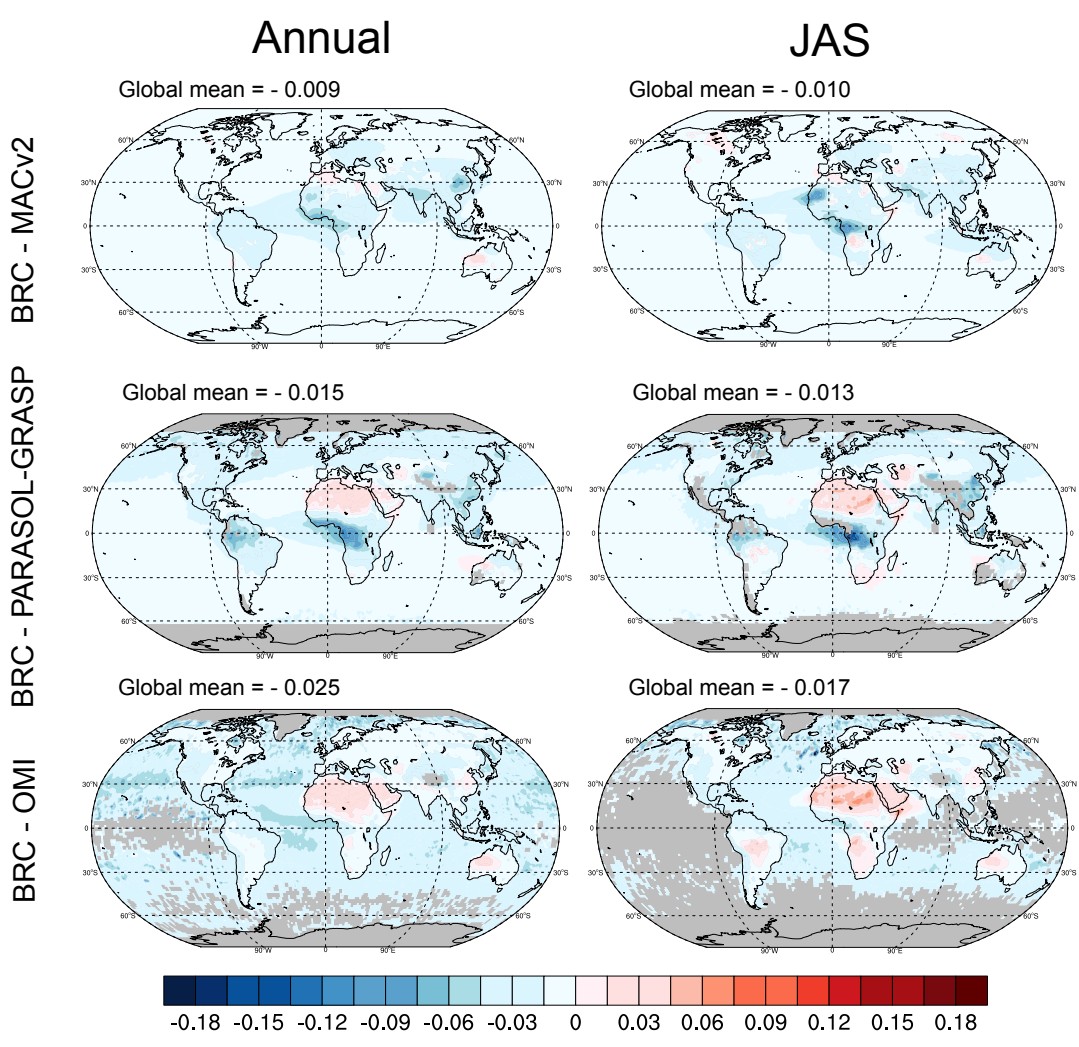

**Figure A3.** Same as Figure A1 but for the AAOD (440 nm, see Tables 2 and 3 for details).