# Peer review of "Modeling radiative and climatic effects of brown carbon aerosols with the ARPEGE-Climat global climate model"

_Atmospheric Chemistry and Physics, 2022_

## Referee Comment (RC1)

**Review of acpd-022-226 manuscript** *'Modeling radiative and climatic effects of brown carbon aerosols with the ARPEGE-Climat global climate model'* by Drugé et al.

**General Overview**

This manuscript describes model-calculated aerosol extinction and absorption optical depth (AOD and AAOD) as well as single scattering albedo (SSA) by the ARPEGE-Climat global climate model. The authors give a detailed description of the modelled processes and parameterizations and evaluate their results by comparison to ground-based and satellite observations.

**General Comments:**

The weakest part of the presented work in the evaluation of their results using satellite observations. For this task they have used the POLDER-GRASP aerosol product as well as an OMI aerosol product.

The accuracy of POLDER-GRASP retrievals has been evaluated by comparisons to AERONET observations as documented in the provided references. As for the OMI aerosol product, the authors seem unaware of the availability of two quite different OMI aerosol products: OMAERO and OMAERUV.

For the OMI-OMAERO satellite product they have used, the authors do not provide any references on the evaluation of AOD and SSA with ground-based observations. Because of the large differences in retrieved aerosol parameters between the two satellite datasets (POLDER-GRASP and OMI-OMAERO), and between the model and OMI-OMAERO as reported in the different tables and figures, especially over the oceans, it is important to properly document the expected accuracy of the reference satellite data sets.

Without a literature-supported accuracy analysis of the OMI-OMAERO aerosol data, this satellite-product should not be used as a reference data set. This issue as well as the minor comments below should be addressed for the paper to be published.

**Specific Comments:**

Pg. 6

Line 29. Although OA's can be considered largely non-absorbing in the visible, they do absorb in the UV.

Pg. 7

Line 9. Add Andreae et al (2019) reference.

Pg. 9

Line 19. The availability of recently produced global representation of spectral aerosol absorption from the combined use of AERONET AOD and satellite radiances from OMI and MODIS observations (Kayetha et al., 2022) should be mentioned.

Line 24. What does 'most satisfactory results' mean?

Line 27. The source of the data shown in Table A1 should be clearly identified. An additional column indicating the source (i.e., reference) should be added.

Line 30. Provide references (or supporting reasoning) for the choice of rain and snow washout efficiencies as well as for BrC fractions content in cloud-mixtures. Please comment on the overall importance of these assumptions (i.e., sensitivity) on the study results.

Pg. 8

Line 2. Provide a literature reported quantitative estimate on *predominance of absorption by primary BrC from BB and BF emissions over that of the absorbing SOA.*

Line 20. The description of the OMI aerosol product is ambiguous. As described in the Torres et al (2007) reference, there are two different aerosol algorithms: OMAERO and OMAERUV. In this work, it looks like the authors refer to the OMAERO product.

Line 22. Remove the Ahmad et al (2003) and Jethva et al (2014) references. The Ahmad et al (2003) reference is a pre-launch publication irrelevant in the context of the work presented here. The Jethva et al (2014) reference addresses the evaluation of the OMAERUV aerosol product.

Line 24. Provide references for the quoted AAOD and SSA uncertainties (0.01 and 0.03) of the OMI-OMAERO aerosol product.

References

Remove Ahmad et al (2003) reference

Remove Jethva et al (2014) reference

References to add:

Andreae, M. O.: Emission of trace gases and aerosols from biomass burning – an updated assessment, Atmos. Chem. Phys., 19, 8523–8546, https://doi.org/10.5194/acp-19-8523-2019, 2019.

Kayetha, V., Torres, O., and Jethva, H.: Retrieval of UV–visible aerosol absorption using AERONET and OMI–MODIS synergy: spatial and temporal variability across major aerosol environments, Atmos. Meas. Tech., 15, 845–877, https://doi.org/10.5194/amt-15-845-2022, 2022.

**Technical Corrections**

Pg. 9

Line 3. Replace OMI with OMI-OMAERO

Line 21. Replace 'including or not' with 'with or without'

Line 23. Replace 'include two members' with 'two modelling configurations'

Pg. 12

Lines 27 and 32.  Replace OMI with OMI-OMAERO

Pg.13

Lines 4, 8, 15, 17. Replace OMI with OMI-OMAERO

Pg. 16

Line 27. Replace OMI with OMI-OMAERO

Pg. 17

Line 32. Replace OMI with OMI-OMAERO

Pg. 29

Replace OMI with OMI-OMAERO in row 1 and caption of Table 2

Pg. 31

Replace OMI with OMI-OMAERO in column 1 row 7 and caption of Table 4

Pg 32

Replace OMI with OMI-OMAERO in column 7 row 1 and caption of Table 5

Pg. 37 to Pg. 44

Replace OMI with OMI-OMAERO in legends and captions of Figures 3 to 10

Appendix

Replace OMI with OMI-OMAERO in legends and captions of Figures A1 to A3

---

## Author Comment (AC1)

We would like to thank the two anonymous reviewers for their comments. We present below our responses to these comments as well as the modifications we made in the paper following these comments.

In order to respond to the different comments and to make the article more robust, we reran our simulations to diagnose the AAOD and SSA of the different aerosols of the TACTIC scheme. Several figures have been added in the revised article. Figure 7 shows the contribution of the model BrC absorption to the total aerosol absorption. In the light of these results we added in our article the evaluation of the aerosol AAOD and SSA at 350 nm (Figures 8 and 10).

**Anonymous Referee 1**

General Overview:
This manuscript describes model-calculated aerosol extinction and absorption optical depth (AOD and AAOD) as well as single scattering albedo (SSA) by the ARPEGE-Climat global climate model. The authors give a detailed description of the modelled processes and parameterizations and evaluate their results by comparison to ground-based and satellite observations.

General Comments:
The weakest part of the presented work in the evaluation of their results using satellite observations. For this task they have used the POLDER-GRASP aerosol product as well as an OMI aerosol product.

The accuracy of POLDER-GRASP retrievals has been evaluated by comparisons to AERONET observations as documented in the provided references. As for the OMI aerosol product, the authors seem unaware of the availability of two quite different OMI aerosol products: OMAERO and OMAERUV.

For the OMI-OMAERO satellite product they have used, the authors do not provide any references on the evaluation of AOD and SSA with ground-based observations. Because of the large differences in retrieved aerosol parameters between the two satellite datasets (POLDER-GRASP and OMI-OMAERO), and between the model and OMI-OMAERO as reported in the different tables and figures, especially over the oceans, it is important to properly document the expected accuracy of the reference satellite data sets.

Without a literature-supported accuracy analysis of the OMI-OMAERO aerosol data, this satellite-product should not be used as a reference data set. This issue as well as the minor comments below should be addressed for the paper to be published.

We have clarified the OMI aerosol product used in this study: it is the OMAERUVd satellite product. For clarity we now use the term OMI-OMAERUVd instead of OMI in the article. Its description has therefore been updated and detailed with relevant references on the evaluation of SSA with ground-based observations (Jethva et al. 2014; Drakousis et al. 2020).

We have added the following paragraph in our manuscript: "- OMI-OMAERUVd (2005-2019, 1°resolution, Torres et al. 2007, 2013) for the AAOD and SSA (350 and 440 nm) and the AOD (550 nm). The OMI (Ozone Monitoring Instrument) OMAERUVd dataset comes from a spectrometer aboard NASA's Earth Observing System's Aura satellite and is archived at the NASA Goddard Earth Sciences Data and Information Services Center. The level 3 daily global gridded product OMAERUVd-v003, used in this study, is produced with all data pixels which fall in a grid box with quality filtered, based on the pixel level OMI Level 2 Aerosol data product OMAERUV. The OMAERUV data product is an improved version of the TOMS version-8 algorithm that essentially uses ultraviolet radiance data (Jethva et al., 2014). The estimated uncertainty in retrieved SSA is of $\pm0.03$ for AOD (440 nm) larger than 0.4. This error is largely attributed to the uncertainty in the instrument calibration (Dubovik et al., 2000; Jethva et al., 2014). AOD over land is expected to have the same root–mean square error (RMSE) as TOMS retrievals (0.1 or 30% whichever is larger). Over ocean, the AOD RMSE is likely to be two times larger. The RMSE for AAOD is estimated to be 0.01 (OMI User's Guide). An evaluation of the OMAERUVd aerosol SSA data through comparisons against daily SSA products from 541 globally distributed AERONET stations for a 15-year period (2005-2019) was carried out in the study of Drakousis et al. (2020). They show that about 50% of OMI-OMAERUVd - AERONET matchups agree within

50 the absolute difference of 0.03 at 440 nm. However, they also indicate that OMI-OMAERUVd tends to overestimate SSA over areas where biomass burning occurs."

Specific Comments:
Pg. 6: Line 29. Although OA's can be considered largely non-absorbing in the visible, they do absorb in the UV. In our BrC
55 parameterization, we separate organic aerosols into aerosols emitted by fossil fuel sources (that we still call organic aerosols or OA) and BrC emitted by biomass burning sources and by biofuel sources. And we make the hypothesis that OA are non absorbing aerosols, which is commonly admitted in modelling studies, and that BrC are absorbing aerosols.

We reformulated our text that now reads :" In this parameterization, BrC corresponds to organic aerosols emitted by BB and
60 BF while OA corresponds to organic aerosols emitted by FF. At this stage, we consider our OA aerosol as a non absorbing aerosol, as shown by most observations (Laskin et al., 2015), while BrC is considered as an absorbing aerosol."

Pg. 7: Line 9. Add Andreae et al (2019) reference. Done.

65 Pg. 9: Line 19. The availability of recently produced global representation of spectral aerosol absorption from the combined use of AERONET AOD and satellite radiances from OMI and MODIS observations (Kayetha et al., 2022) should be mentioned. A sentence has been added in the revised version: "It can be noted that a global representation of the spectral aerosol absorption in the UV-to-visible wavelength range (340–670 nm) based on a synergy of ground measurements (AERONET AOD) and of satellite observations (near-UV OMI radiances and visible MODIS (Moderate Resolution Imaging Spectroradiomete)
70 radiances) is presented in Kayetha et al. (2022)."

Pg. 7: Line 24. What does 'most satisfactory results' mean? There are few studies that investigate the level of BrC absorption decrease over time (bleaching effect). To study this parameter, we carried out 3 tests with several absorption decreases (25, 50 and 75% after one day). Pending further studies on this subject, we decided to take an average BrC absorption decrease over
75 time, 50% after one day. This average value correspond in our sensitivity tests to the most realistic results when compared to our reference datasets.

We changed our text that now reads: "The best comparison with our reference data sets was obtained with the 50% value. For clarity reasons..."
80
Pg. 7: Line 27. The source of the data shown in Table A1 should be clearly identified. An additional column indicating the source (i.e., reference) should be added. An additional table (Table A2) with the reference for the refractive index of each aerosol type used in the Mie code to compute optical properties has been added.

85 Pg. 7: Line 30. Provide references (or supporting reasoning) for the choice of rain and snow washout efficiencies as well as for BrC fractions content in cloud-mixtures. Please comment on the overall importance of these assumptions (i.e., sensitivity) on the study results. This sensitivity issue is an interesting one that has not been addressed in this work. Due to the lack of data in literature and for consistency reasons we decided to parameterise BrC deposition (wet and dry) as OA deposition. Indeed, we made this choice in order to observe impacts largely related to changes in optical properties. References for the rain and snow
90 washout efficiencies as well as for the BrC fraction content in cloud-mixtures are Michou et al. (2015) and Bourgeois and Bey (2011). These references have been added in the revised version.

Pg. 8: Line 2. Provide a literature reported quantitative estimate on predominance of absorption by primary BrC from BB and BF emissions over that of the absorbing SOA. In their study, Wang et al. 2016 indicate "As the absorption from primary OA
95 (Br-POA) from biofuel and biomass burning typically dominates that of absorbing SOA (Br-SOA) (Martinsson et al., 2015; Laskin et al., 2015), the absorption of Br-SOA is much more challenging to detect than Br-POA in most field measurements." Furthermore, Saleh et al. 2014 say "SOA is less absorptive than primary OA, but has a stronger wavelength dependence". In a previous study (Saleh et al. 2013) state: "For the investigated fuels, SOA is less absorptive than POA in the long visible,

but exhibits stronger wavelength-dependence and is more absorptive in the short visible and near-UV." The study of Kumar et al. 2018 also indicates: "The corresponding mass absorption cross section of POA (5.5 m2 g-1) was higher than that of SOA (2.4 m2 g-1) at 370 nm. However, SOA presents a substantial mass fraction, with a measured average SOA / POA mass ratio after aging of $\sim 5$ and therefore contributes significantly to the overall light absorption, highlighting the importance of wood-combustion SOA as a source of atmospheric brown carbon."

On the basis of the above literature, we have modified the sentence of the article: "One limitation of this study is to neglect absorption by biogenic (Lin et al., 2014; Saleh et al., 2015) and aromatic SOA (Wang et al., 2014; Jo et al., 2016; Wang et al., 2018). Some studies show that the absorption of the primary BrC from BB and BF emissions usually dominates that of the absorbing SOA (Saleh et al., 2013; Martinsson et al., 2015; Wang et al., 2016). However, Kumar et al. (2018) indicate in their study that SOA, after aging, can contribute significantly to the overall absorption."

Pg. 8: Line 20. The description of the OMI aerosol product is ambiguous. As described in the Torres et al (2007) reference, there are two different aerosol algorithms: OMAERO and OMAERUV. In this work, it looks like the authors refer to the OMAERO product. Thank your for asking for clarifications on the OMI aerosol product. We double checked what we used, and in this work, we have used OMAERUVd. The description of the OMI aerosol product has therefore been clarified and detailed with relevant references.

Pg. 8: Line 22. Remove the Ahmad et al (2003) and Jethva et al (2014) references. The Ahmad et al (2003) reference is a pre-launch publication irrelevant in the context of the work presented here. The Jethva et al (2014) reference addresses the evaluation of the OMAERUV aerosol product. The Ahmad et al (2003) reference has been removed. On the other hand, as we use OMI-OMAERUVd, the reference Jethva et al (2014) has been kept.

Pg. 8: Line 24. Provide references for the quoted AAOD and SSA uncertainties (0.01 and 0.03) of the OMI-OMAERO aerosol product. Done. References for the SSA are Dubovik et al (2000) and Jethva et al (2014). For the AOD and the AAOD the information comes from the OMI User's Guide.

References
Remove Ahmad et al (2003) reference
Remove Jethva et al (2014) reference
References to add:
Andreae, M. O.: Emission of trace gases and aerosols from biomass burning – an updated assessment, Atmos. Chem. Phys., 19, 8523–8546, https://doi.org/10.5194/acp-19-8523-2019, 2019.
Kayetha, V., Torres, O., and Jethva, H.: Retrieval of UV–visible aerosol absorption using AERONET and OMI–MODIS synergy: spatial and temporal variability across major aerosol environments, Atmos. Meas. Tech., 15, 845–877, https://doi.org/10.5194/amt-15-845-2022, 2022.
Done, except for the Jethva et al (2014) reference which is appropriate to the OMAERUV aerosol product.

Technical Corrections:
Pg. 9: Line 3. Replace OMI with OMI-OMAERO
Done with OMI-OMAERUVd.

Pg. 9: Line 21. Replace 'including or not' with 'with or without' Done.

Pg. 9: Line 23. Replace 'include two members' with 'two modelling configurations' We have rephrased our sentence: "All simulations consist in 30-year AMIP-type simulations with prescribed monthly sea surface temperature (SST) and sea ice fraction. The period covered is 2000-2014, it is simulated twice for each simulation (by changing the initial state of the atmosphere), so the total number of simulated years is of 30."

Pg. 12: Lines 27 and 32. Replace OMI with OMI-OMAERO Done with OMI-OMAERUVd.

Pg.13: Lines 4, 8, 15, 17. Replace OMI with OMI-OMAERO Done with OMI-OMAERUVd.

Pg. 16: Line 27. Replace OMI with OMI-OMAERO Done with OMI-OMAERUVd.

Pg. 17: Line 32. Replace OMI with OMI-OMAERO Done with OMI-OMAERUVd.

Pg. 29: Replace OMI with OMI-OMAERO in row 1 and caption of Table 2 Done with OMI-OMAERUVd.

Pg. 31: Replace OMI with OMI-OMAERO in column 1 row 7 and caption of Table 4 Done with OMI-OMAERUVd.

Pg 32: Replace OMI with OMI-OMAERO in column 7 row 1 and caption of Table 5 Done with OMI-OMAERUVd.

Pg. 37 to Pg. 44: Replace OMI with OMI-OMAERO in legends and captions of Figures 3 to 10 Done with OMI-OMAERUVd.

Appendix: Replace OMI with OMI-OMAERO in legends and captions of Figures A1 to A3 Done with OMI-OMAERUVd.

---

## Author Comment (AC2)

We would like to thank the two anonymous reviewers for their comments. We present below our responses to these comments as well as the modifications we made in the paper following these comments.

170 In order to respond to the different comments and to make the article more robust, we reran our simulations to diagnose the AAOD and SSA of the different aerosols of the TACTIC scheme. Several figures have been added in the revised article. Figure 7 shows the contribution of the model BrC absorption to the total aerosol absorption. In the light of these results we added in our article the evaluation of the aerosol AAOD and SSA at 350 nm (Figures 8 and 10).

**Anonymous Referee 2**

175

This manuscript presents a global model simulation of brown carbon and its climate effect in the ARPEGE-Climat global climate model. The authors provided a detail review of current modeling schemes for BrC and a detail introduction to their own parameterizations. They evaluated the simulated AOD, SSA, AAOD with satellite and ground-based sun-photometer measurements.

180

The biggest issue of this manuscript is the model-observation comparison part, which in my mind, doesn't provide much useful information. The comparisons use AOD, SSA, and AAOD at 440nm as well as AOD at 550nm. Since BrC could contribute only a small part to those total aerosol properties, the model biases are more likely to be related to other aerosols: We agree that this model-observation comparison part needed improvement. First, we reran our simulations in order to diagnose the AAOD

185 and SSA of the different aerosols of the TACTIC scheme at short wavelengths (350 and 440 nm). With these diagnostics, we produced a new figure that appears as Figure 7 in the revised article. This figure shows how the model BrC absorption contributes to the total aerosol absorption, and in the light of these results we added in our article the evaluation of the aerosol AAOD and SSA at 350 nm. For doing that, we processed additional reference datasets, namely the OMI-OMAERUVd and MACv2 products. Concerning the AOD at 550 nm, it is tuned in each simulation to be as consistent as possible with the merged

190 AOD product FMI_SAT (see text of the article for details). So its evaluation is of less interest and figures have therefore been placed in Appendix.

- At 550nm, BrC absorption is too small to significantly affect total AOD. In contrast, the uncertainty of other aerosols in the simulation are much larger due to the assumptions such as: no anthropogenic SOA is considered, applying constant scale

195 factors for emissions globally, etc. We agree that conclusions of our analysis could be quite different with different assumptions we could make in our TACTIC aerosol scheme. It would certainly be of great interest to the aerosol community to know about the relative importance of various assumptions. This relative importance is of course model-dependant, and model-objective dependant. So the work is quite immense. We chose here to evaluate addition of the BrC species keeping the rest of the model (which has been evaluated in various contexts, see references in the article) unchanged. As previously mentioned, the AOD

200 at 550 nm is tuned in each simulation to be as consistent as possible with the merged AOD product FMI_SAT. The different corresponding figures have been placed in appendix (Figures A1, A2, A5 and A6) to focus our evaluation on the aerosols SSA and AAOD at 350 and 440 nm (see also above our response to the previous comment).

- Even for AAOD at 440nm, black carbon usually contributes more absorption than BrC. An evaluation of BC AAOD is needed

205 (at least based on previous literature) and its influence on the model-observation comparison should be discussed. In order to provide as much information as possible on this subject, we have added in the introduction several references comparing the absorption of BrC to BC at different wavelengths:

"Different studies highlighted that the BrC absorption is not negligible, and is even comparable to that of BC at short wave-

210 lengths (Alexander et al., 2008; Bahadur et al., 2012; Chung et al., 2012; Kirchstetter and Thatcher, 2012; Pokhrel et al., 2017). Using aerosol optical properties derived from Aerosol Robotic Network (AERONET) measurements, Bahadur et al. (2012) estimated that BrC absorption at 440 nm could be about 40% of BC absorption. On the other hand they also showed, at 675 nm, that BrC absorption is less than 10% of BC absorption. Kirchstetter and Thatcher (2012), using residential wood smoke samples, found that BrC absorption contributes 49% of carbonaceous aerosols (BC + OA) absorption at wavelengths below

215 400 nm. Lastly, based on laboratory measurements using a multi-channel photoacoustic absorption spectrometer, Pokhrel et al. (2017) showed that BrC absorption at shorter visible wavelengths is of equal or greater importance to that of BC, with respectively maximum contributions of up to 92% and 58% of total aerosol absorption at 405 and 532 nm."

220 To our knowledge, there are no data available to evaluate our BC AAOD only. On the other hand, in the "Model results" section, Figure 7 has been added to provide quantitative information on the BrC absorption compared to that of BC in our model at 350, 440 and 550 nm. We added the following sentence in our article:

"This figure shows an important BrC absorption, comparable to that of BC at short wavelengths, especially during the JAS period. In details, BrC absorption is about 45% of BC absorption at 350 nm and 35% at 440 nm. These results are consistent
225 with the study of Bahadur et al. (2012) that estimates a BrC absorption at 440 nm of about 40% of BC absorption. At 550 nm, Figure 7 shows a lower BrC absorption, which is less than 30% of BC absorption. For comparison with other studies, Samset et al. (2018) show with the LMDZ-INCA model a simulated BrC AAOD at 550 nm of about 20% of that of BC. In this section we will therefore evaluate the aerosol scheme at 350 and 440 nm, which are the wavelengths where BrC contributes the most to the total absorption."

230

In addition, the authors should be clear about the datasets they used:

- How are the SSA and AOD retrieved in the satellite products? Are the retrievals including any assumptions? Are those assumptions consistent with those in your model? Thank you for these questions. A reference is provided for each satellite
235 product or reference dataset indicating their main characteristics and the method used. In addition, the uncertainties on parameters studied in the article are indicated for each product if available. We also indicate in the article the assumptions we could identify in the documentation of the reference products we used. In our model we can output some diagnostics that really comply to satellite observations. These diagnostics come from the CFMIP (Cloud Feedback Model Intercomparison Project) Observation Simulator Package (COSP), embedded in the CNRM model. However this version of the COSP package does not
240 output model diagnostics for satellite products we are interested in this study (OMI, PARASOL-GRASP). The evaluation we made can thus be improved.

- The OMI observations used here was in fact the OMAERO product, please clarify. On the other hand, OMAERUV may be a better choice for this work. We have clarified the OMI aerosol product used in this study: it is the OMAERUVd satellite
245 product. Its description has therefore been updated and detailed with appropriate references. For clarity reasons we now use the term OMI-OMAERUVd instead of OMI.

- For AERONET, level 2.0 data were used, which only have SSA and AAOD information when AOD>=0.4. Therefore, monthly average values are not able to represent the real monthly condition and tend to overemphasize the high AOD hours. They are
250 not appropriate to be compared with monthly values from the model. In order to compare the AERONET data and our model results, we finally used the AERONET level 1.5 data. Indeed, unlike level 2.0, level 1.5 reports SSA also for AOD 440nm $\leq$ 0.4 so these data appeared to us more appropriate to be compared with our model data. The description of the AERONET data has therefore been changed in the article.

255 Specific comments:

p.4, line 31: Could you please provide which six bands are used in shortwave? This could be important as BrC absorption is very sensitive to spectral bands. The limits of six spectral bands used in the shortwave radiation scheme are now listed in the article (0.185, 0.25, 0.44, 0.69, 1.19, 2.38 and 4.00 microns).

260

p.7, line 15: Please provide justification or reference for BrC size distribution. GMD of 100nm looks small for biofuel and biomass burning organics. Thank you for noting that. This was a mistake. 100 nm is the radius and not the diameter. The BrC geometric median diameter has therefore been corrected in the text. We have also added references for the BrC size distribution

**6**

(GMD and standard deviation):

"The BrC geometric median diameter is assumed to be of 0.2 µm (Saleh et al., 2015) with a standard deviation of 1.6 (Wang et al., 2018; Tuccella et al., 2020)."

p.7, line 18: Does it mean you assume all the freshly emitted OA from biofuel and biomass burning are hydrophobic? Is there any difference between your hydrophilic and hydrophobic BrC, other than the optical properties? Are treatments in BRC and NOBRC simulations same? The freshly emitted OA (BB+BF+FF sources in NOBRC and FF source in BRC) is treated in the same way in the NOBRC and BRC simulations (50% hydrophobic and 50% hydrophilic with a characteristic time of 1.63 day for the passage hydrophobic-hydrophilic). In the BRC simulation, all the freshly emitted brown carbon (BB+BF sources) is considered hydrophobic (with a characteristic time of 1 day for the passage hydrophobic-hydrophilic). Except for the optical properties and the scavenging there are no differences between our hydrophilic and hydrophobic BrC.

p.7 line 11: I assume you mean "not all of the burning conditions are represented" for "all burning conditions are not represented". Done.

p.9, line 21: Better use "with or without" for "including or not" Done.

p.9, line 23: What do you mean for "two members"? We meant that the covered period (2000-2014) was simulated twice for each simulation by changing the initial state of the atmosphere. We have rephrased our sentence in the revised version: "All simulations consist in 30-year AMIP-type simulations with prescribed monthly sea surface temperature (SST) and sea ice fraction. The period covered is 2000-2014, it is simulated twice for each simulation (by changing the initial state of the atmosphere), so the total number of simulated years is of 30."

p.13, line 14: Change "Our last comparisons concerns" to "Our last comparison concerns" Done.

p.17, line 17: Change "non only" to "not only". There are many grammar errors are indicated in the above comments. Please check your writing carefully. Done.